# The androgen receptor is a therapeutic target in desmoplastic small round cell sarcoma

Salah-Eddine Lamhamedi-Cherradi [1,12,13✉], Mayinuer Maitituoheti[2,12], Brian A. Menegaz [3], Sandhya Krishnan [1], Amelia M. Vetter[1], Pamela Camacho[4], Chia-Chin Wu [2], Hannah C. Beird [2], Robert W. Porter [1], Davis R. Ingram[5], Vandhana Ramamoorthy[2], Sana Mohiuddin[6], David McCall [6], Danh D. Truong [1], Branko Cuglievan[6], P. Andrew Futreal [5], Alejandra Ruiz Velasco[1], Nazanin Esmaeili Anvar [5], Budi Utama[7], Mark Titus[8], Alexander J. Lazar [5], Wei-Lien Wang [5], Cristian Rodriguez-Aguayo [9], Ravin Ratan [1], J. Andrew Livingston [1], Kunal Rai [2,13✉], A. Robert MacLeod[10], Najat C. Daw [6], Andrea Hayes-Jordan[11] & Joseph A. Ludwig [1,13✉]

Desmoplastic small round cell tumor (DSRCT) is an aggressive, usually incurable sarcoma subtype that predominantly occurs in post-pubertal young males. Recent evidence suggests that the androgen receptor (AR) can promote tumor progression in DSRCTs. However, the mechanism of AR-induced oncogenic stimulation remains undetermined. Herein, we demonstrate that enzalutamide and AR-directed antisense oligonucleotides (AR-ASO) block 5α-dihydrotestosterone (DHT)-induced DSRCT cell proliferation and reduce xenograft tumor burden. Gene expression analysis and chromatin immunoprecipitation sequencing (ChIP-seq) were performed to elucidate how AR signaling regulates cellular epigenetic programs. Remarkably, ChIP-seq revealed novel DSRCT-specific AR DNA binding sites adjacent to key oncogenic regulators, including WT1 (the C-terminal partner of the pathognomonic fusion protein) and FOXF1. Additionally, AR occupied enhancer sites that regulate the Wnt pathway, neural differentiation, and embryonic organ development, implicating AR in dysfunctional cell lineage commitment. Our findings have direct clinical implications given the widespread availability of FDA-approved androgen-targeted agents used for prostate cancer.

[1] Sarcoma Medical Oncology Department, The University of Texas MD Anderson Cancer Center, Houston, TX 77030, USA. [2] Department of Genomic Medicine, The University of Texas MD Anderson Cancer Center, Houston, TX 77030, USA. [3] Department of Surgery, Breast surgical Oncology, Baylor College of Medicine, Houston, TX 77030, USA. [4] Texas Children's Cancer & Hematology Centers, Houston, TX 77384, USA. [5] Division of Pathology, The University of Texas MD Anderson Cancer Center, Houston, TX 77030, USA. [6] Division of Pediatrics, The University of Texas MD Anderson Cancer Center, Houston, TX 77030, USA. [7] Optical Microscopy Facility, Rice University, Houston, TX 77030, USA. [8] Genitourinary Medical Oncology Department, The University of Texas MD Anderson Cancer Center, Houston, TX 77030, USA. [9] Experimental Therapeutics Department, The University of Texas MD Anderson Cancer Center, Houston, TX 77030, USA. [10] Ionis Pharmaceuticals, Carlsbad, CA 92010, USA. [11] Lineberger Comprehensive Cancer Center, UNC, Chapel Hill, NC 27599, USA. [12] These authors contributed equally: Salah-Eddine Lamhamedi-Cherradi, Mayinuer Maitituoheti. [13] These authors jointly supervised this work: Salah-Eddine Lamhamedi-Cherradi, Kunal Rai, Joseph A. Ludwig. ✉email: SLamhamedi@mdanderson.org; Krai@mdanderson.org; jaludwig@mdanderson.org

Desmoplastic small round cell tumor (DSRCT) is an aggressive soft-tissue malignancy that usually presents in post-pubertal adolescents and young adults as a large intra-abdominal mass together with a widespread coating of serosal and subdiaphragmatic surfaces by hundreds to thousands of malignant nodules. Given the inconspicuous tumor location, nearly all patients present in an advanced stage with abdominal pain or distention, nausea, constipation, and weight loss[1].

DSRCT's cell of origin is unknown, and these tumors universally exhibit a high-grade, poorly differentiated state characterized by cellular nests of unclear lineage, seeming to show epithelial, muscular, mesenchymal, and neural differentiation admixed with prominent desmoplastic stroma[2–4]. Given its rarity, with an age-adjusted incidence peak incidence of 0.3–0.74 cases per million[5], it was not until 1989 that Gerald and Rosai first described DSRCT as a unique clinicopathologic disease[6]. Shortly thereafter, cytogenetic analyses demonstrated that DSRCT tumors harbor a pathognomonic t(11;22)(p13:q12) chromosomal translocation that pairs the Ewing sarcoma (ES) gene (*EWSR1*) with the Wilms tumor suppressor gene (*WT1*)[6–9]. The resulting chimeric 59 kDa fusion protein (FP), in concert with the heterozygous functional loss of the WT1 tumor suppressor protein, promotes an oncogenic effect that reinforces tumor survival and growth[10].

The PI3K/AKT and androgen receptor (AR) signaling cascades are among the most frequently activated in cancer[11,12]. Given the striking observation that 90% of DSRCT cases occur in post-pubertal males (with an average age at diagnosis of 21.4 years), we investigated how AR contributes to tumorigenesis and survival[4,13,14]. A potential connection between the AR and DSRCT, first reported by Fine et al. in 2006, studied a series of twenty-seven advanced-stage DSRCT patients who had progressed through at least two chemotherapy regimens[15]. In that retrospective multi-center analysis, 37% of the samples were ≥2+ by immunohistochemistry, and surprisingly, three of six AR+ patients transiently benefited from second-generation combined androgen blockade (CAB) using Lupron and bicalutamide. Despite that promising signal of activity, AR targeting has not been pursued further in the clinic, and at the time of this writing, no additional studies were reported in the literature.

In the present work, we use DSRCT xenografts and patient-derived tumor explants (PDXs) to extend the findings by Fine et al. to modern-day AR-targeted therapies, such as enzalutamide, that form the backbone of prostate cancer (PC) treatment[16]. In addition, we present promising efficacy data using an experimental AR-targeted antisense oligonucleotide (ASO) that significantly delayed tumor growth by suppressing AR expression. Finally, we present chromatin immunoprecipitation sequencing (ChIP-seq) results that suggest a new mechanistic understanding of AR's role in DSRCT tumorgenicity. Our research findings substantiate DSRCT as a second AR-driven malignancy and implicitly suggest a path toward clinical trials that center on AR-directed treatment options for this otherwise intractable pediatric cancer.

## Results

**Protein expression in DSRCT differs substantially from ES.** Given the differences in clinical presentation, tumor biology, and response to biologically targeted therapies, we conducted a reverse-phase protein lysate array (RPPA) to identify proteins enriched in DSRCT. To determine tumor-specific proteins, we compared protein lysates from DSRCT nodules and paired adjacent normal-appearing mesenteric tissue from the same patients using a well-described RPPA platform enriched for known oncoproteins (Supplementary Fig. S1)[17–19]. Unsupervised

double-hierarchical clustering correctly separated normal mesenteric tissues from DSRCT, which overexpressed Akt, Syk, PKC-α, and other proteins. Next, we identified proteins enriched in DSRCT compared to another malignancy, using ES as the closest molecular cousin that shares an N-terminus EWSR1 fusion partner. Interestingly, AR and Syk proteins were upregulated in most DSRCT specimens but nearly undetectable in ES (Fig. 1a, b). The RPPA data, validated by western blots (WB), provides the first screen of proteins enriched in DSRCT (Fig. 1c, d).

**Expression of AR in DSRCT primary tumors.** Since AR activity requires androgen-mediated nuclear translocation, we created a DSRCT-specific tissue microarray (TMA) from 60 cases treated at MD Anderson Cancer Center (MDACC) to determine the prevalence and cellular distribution of AR staining in patients treated at a single institution. Seventy-five percent of the cores available for analysis were positive for nuclear AR by immunohistochemistry (Fig. 2a, b and Supplementary Fig. S2a–c), usually more prominently in the epithelioid cells rather than the desmoplastic stromal cells. Of the AR-positive samples, 7% showed focal AR expression in 10–50% rare, scattered cells and could be of any intensity (low positive AR expression), 3.3% had low AR expression where only 1–10% of cells are positive for AR expression (focal expression), and 25% demonstrated negative AR expression (where only 0–1% of cells are positive). 65% showed high AR expression, defined by AR-positivity in >50% of a sample's cells (Fig. 2b and Supplementary Fig. S2c). High-intensity staining was defined as completely obscuring the nuclear hematoxylin counterstain, while moderate staining allowed visualization of the stain. Weak staining required examination of the cells at least 200× to detect staining reliably. As an additional metric of AR expression, we evaluated an additional 12 DSRCT patient tumors for AR expression by western blotting: 42% of the tumors showed high AR expression, 33% had moderate AR expression, and 25% were AR-negative (Fig. 2c, d). As IHC and western blotting revealed moderate to high AR expression in ~three-quarters of the DSRCT cases assessed, this seemed to substantiate the RPPA results.

**Genomic profiling of DSRCT, PC, and other sarcoma primary tumor samples.** Given the male predominance of DSRCT, new AR protein expression data, and known abundance of AR-targeted therapies used for PC, we elected to pursue AR as a potential therapeutic target in DSRCT. The case report by Fine et al. hinted that DSRCT patients can respond briefly to a first-generation CAB, which heightened our enthusiasm to investigate how AR signaling contributes to DSRCT biology[15].

Given the AR's central role in PC growth and survival, and deep mechanistic understanding of that malignancy[20–23], we performed a gene expression analysis comparing 22 DSRCT samples to 12 PC samples and a group of other diverse sarcoma subtypes, including 7 chondrosarcomas, 7 well-differentiated liposarcomas, 10 dedifferentiated liposarcoma, and 47 osteosarcoma samples, which served as negative controls. As expected, DSRCT demonstrated significant AR upregulation compared to these other sarcoma samples from chondrosarcoma, well-differentiated and dedifferentiated liposarcoma, and osteosarcoma ($p < 0.01$–$p < 0.0001$) but did not surpass the levels observed in PC (Fig. 2e, f).

Compared to other sarcoma subtypes, most DSRCT samples clustered together based upon their expression of 89 genes associated with the androgen pathway, as defined by KEGG (Supplementary Fig. S2d). Similarly, DSRCT samples clustered together by their expression of 55 genes linked to canonical

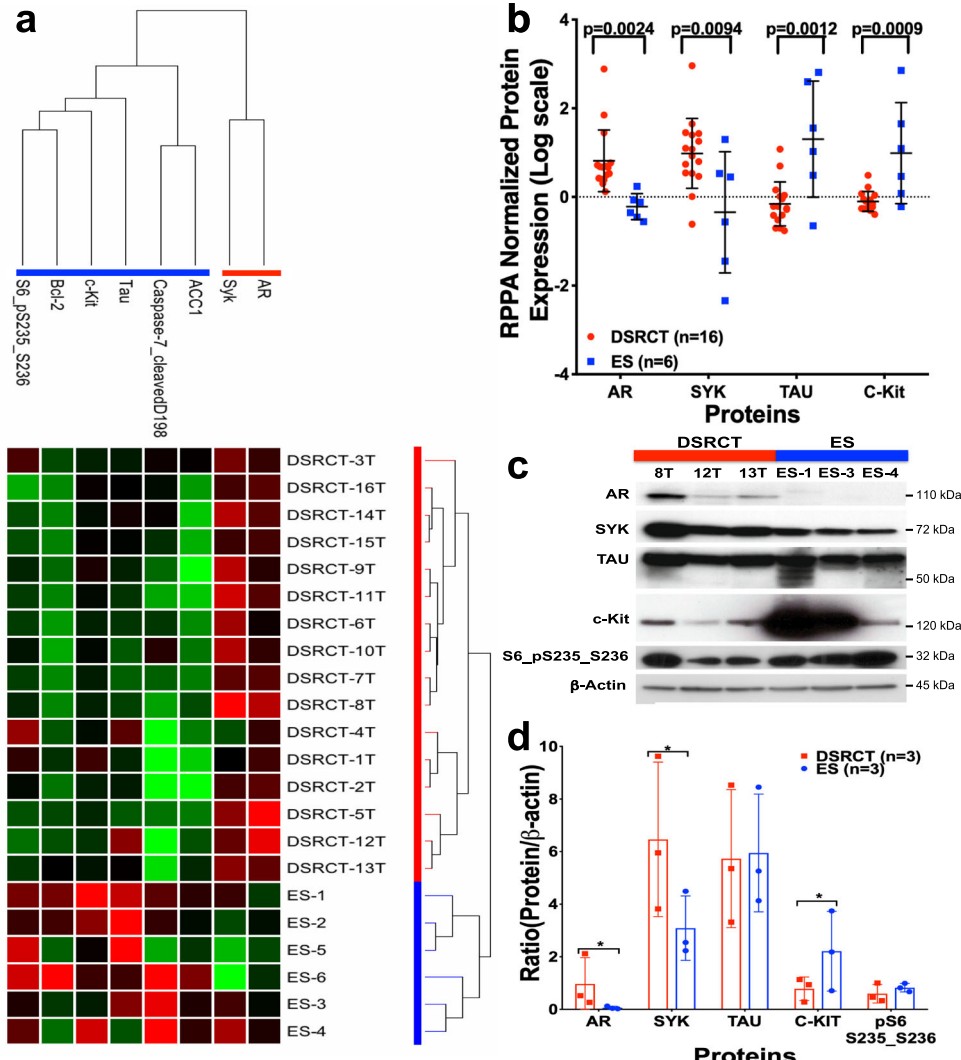

**Fig. 1 Proteomic comparison of DSRCT and ES. a** The protein lysates from DSRCT (red; $n = 16$) and ES (blue; $n = 6$) were subjected to RPPA analysis for 151 proteins and phosphoproteins (red, increased signal; green, decreased signal). Unsupervised double-hierarchical clustering using the Pearson correlation distance metric between proteins (rows) and Centroid linkage (a clustering method) separated the 22 samples into two groups by tumor type (columns). Of the 22 proteins, 8 had expression that differed significantly between ES and DSRCT ($p \leq 0.05$; fold-change $\geq 2$). **b** The mean expression intensity values of the 8 proteins associated with DSRCT or ES and their statistical significance after normalization for global protein expression by median centering across 151 antibodies in the RPPA panel. **c** Western blotting was used to validate the proteins identified by RPPA as being differentially expressed between DSRCT and ES. **d** Normalized protein expression is relative to β-actin. Data points in **b** and **d** represent mean ± SD. $n$ is the number of samples analyzed for each sarcoma subtype.

androgen signaling (Supplementary Fig. S2f). Notably, PC and DSRCT clustered together on their own branch, apart from other sarcoma subtypes, based upon the 1500 most varied genes among all samples (Fig. 3). Pathway analysis revealed other cancer pathways enriched in DSRCT compared to other sarcoma subtypes. This included the cell-adhesion molecule communication pathway that regulates cell invasion through a coordinated balance between adhesion and detachment of cells[24,25] and the extracellular matrix (ECM) interaction pathway, known to initiate cell motility across the ECM barrier[26].

**Characterizing extranuclear partners of AR and its nuclear cofactors.** AR splice variants (AR-Vs) have been implicated in PC tumor progression, an increased risk of biochemical relapse, and inferior overall survival outcomes[27–29]. To determine if variant forms of AR exist in DSRCT, we assessed AR-V7 expression within a cohort of twelve consecutive DSRCT patients. As none of

the initial twelve samples expressed AR-V by IHC, we elected not to examine this further in the broader sample set.

As AR activity can be influenced by integrins and transcriptional co-regulators[30,31], we evaluated these AR activators within the same cohort of 11 DSRCT tumors at the proteomic level using a western blotting analysis. Among three alpha integrin (ITGAV, ITGA4, and ITGA5) and beta integrin (ITGB1, ITGB3, and ITGB5) subunits commonly observed in mesenchymal tissues, protein expression varied considerably and did not correlate with AR expression (Supplementary Fig. S3a).

Since the epigenetic effects of AR can be modified by cofactor binding and matrix metalloproteins, we assessed whether steroid receptor coactivators NCOA1/2/3 or MMP2/13 contribute to the development of DSRCT through AR-dependent mechanisms[32–37]. To accomplish this, we performed a WB of 11 primary DSRCT tumors and the JN-DSRCT and LNCaP PC cell lines. The three-NCOA biomarkers demonstrated heterogeneous expression in the DSRCT clinical samples proportional to their AR expression

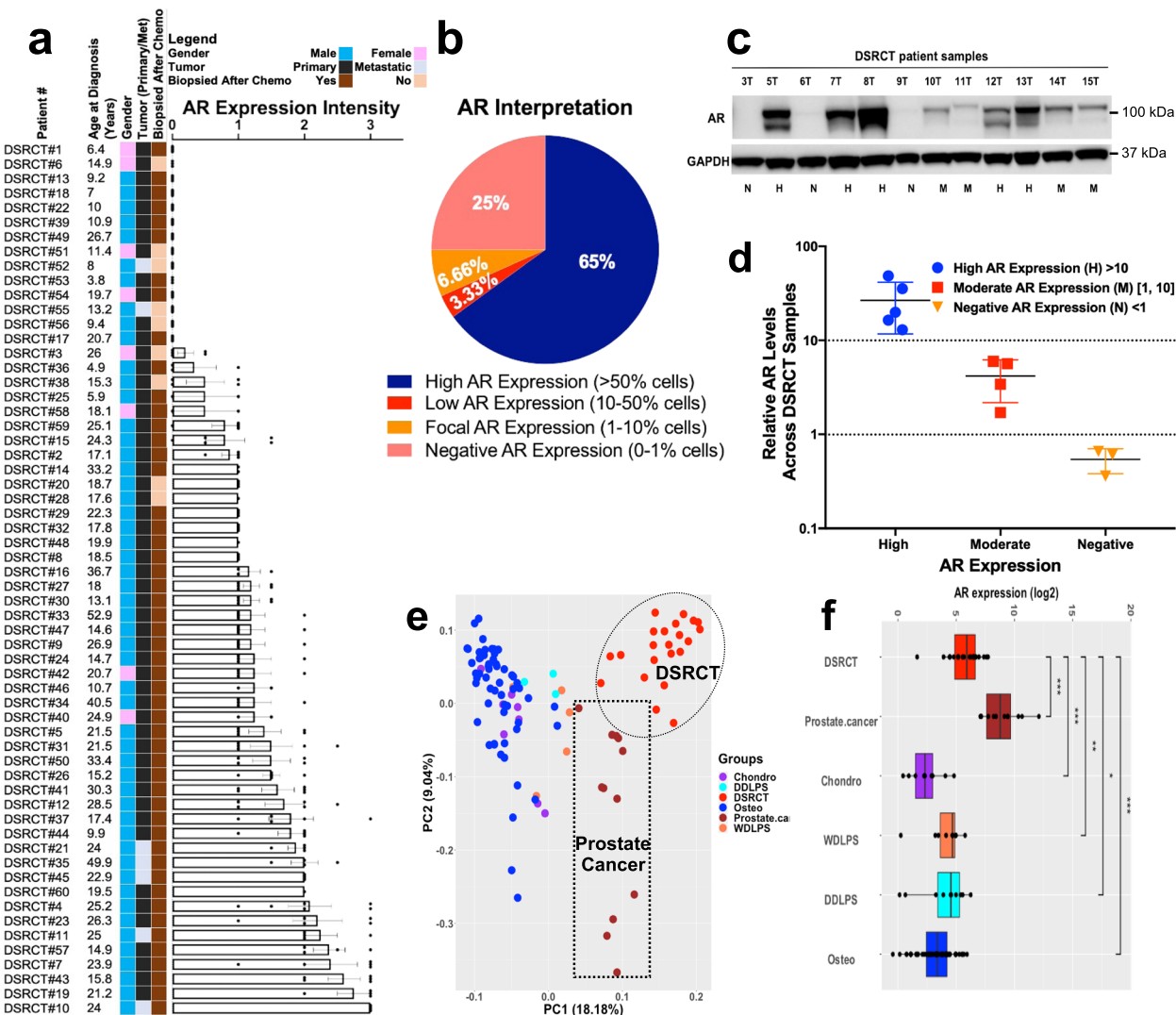

**Fig. 2 DSRCT TMA and frozen specimen profiling for AR and PSA expression. a** A histogram showing the AR IHC expression levels of 60 human DSRCT tumors grouped by intensity (low, moderate, and high). The demographic data, including the corresponding gender (red: male or green: female), the age at diagnosis, and the pre/post-chemotherapy treatment to the surgery of each primary or metastatic resected tumor patient, are displayed at the left of each histogram. **b** AR expression level interpretation on DSRCT TMA IHC-stain and percentage scoring of tumoral labeling (positive (>50%), low positive (10–50%), focal (1–10%), and negative (0–1%)). **c** Western blotting analyses of AR expression in 11 DSRCT snap-frozen primary tumors. AR expression: P = positive, N = negative, or M = moderate. **d** Relative AR levels across samples shown in **c**. Bars show mean ± SD. **e** The principal components analysis plot performed on gene expression from prostate cancer (PC), DSRCT, and additional type of sarcomas samples. **f** Boxplot for the AR gene expression level across DSRCT, prostate cancer, and four other sarcoma types. The Wilcoxon rank-sum test performed to compare the AR levels between DSRCT ($n = 22$) and each of the other cancer types. PC = prostate cancer ($n = 12$); CS = chondrosarcoma ($n = 7$); OS = osteosarcoma ($n = 47$); WDLPS = well-differentiated liposarcoma ($n = 7$), and DDLPS = dedifferentiated liposarcoma ($n = 10$). ***$p$ value < 0.001, **$p$ value < 0.01, and *$p$ value < 0.05.

(Supplementary Fig. S3b). However, the JN-DSRCT cells showed low expression of NCOA1/2 and equivalent expression of NCOA2 versus LNCaP PC cells (Supplementary Fig. S3c). Further investigation with a larger sample set will be required to determine how the AR-dependent integrin/NCOA-dependent pathway impacts DSRCT cell migration and death.

**In vitro stimulation and inhibition of DSRCT proliferation via AR.** Though AR activation by testosterone and DHT leads to brisk PC cell proliferation[38], it was uncertain whether DSRCT cells similarly relied upon AR signaling for proliferation, growth, and survival. To evaluate this, we performed in vitro cell proliferation assays following DHT-mediated AR stimulation in JN-DSRCT; AR-expressing LNCaP PC cells, AR-non-expressing

PC3 PC cells, and ES TC71 cells that were used as positive or negative controls. As hypothesized, DHT stimulation increased cell proliferation of JN-DSRCT and LNCaP cells compared to PC3 and ES cells (Fig. 4a). As measured by western blotting, we confirmed strong AR expression by LNCaP and JN-DSRCT cell lines following DHT stimulation in contrast to its absence in the TC71 ES and PC3 PC cells (Fig. 4b). Next, we performed confocal immunofluorescence staining of these cells to determine if (and how quickly) DHT-mediated stimulation would facilitate AR transmigration from the cytoplasm into the cell nucleus. Our results suggest that AR-nuclear translocation begins within 5 h of DHT exposure and peaks within 24 h (Fig. 4c, d).

Having shown that DHT stimulates DSRCT cells, we explored whether FDA-approved and experimental AR antagonists had an antiproliferative effect. Both enzalutamide (Fig. 4e) and the novel

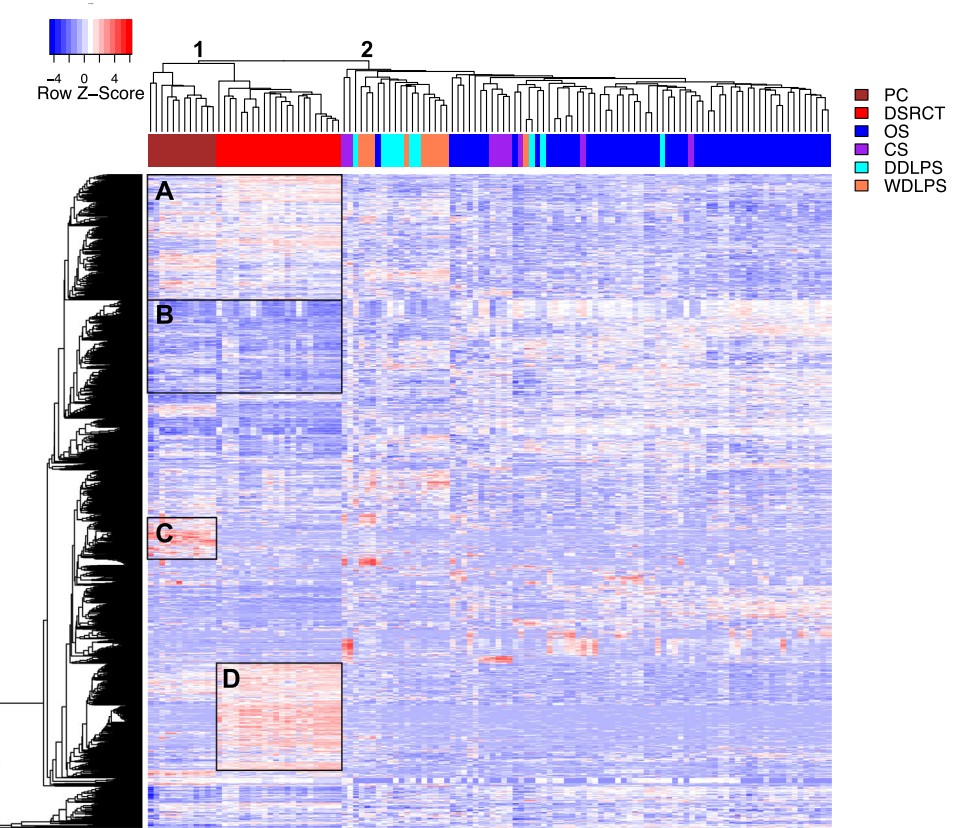

**Fig. 3 Double-hierarchical clustering of PC (n = 12), DSRCT (n = 22), and other sarcoma subtypes (n = 71).** The top 1500 most variable genes across all samples were used to compare DSRCT to PC and other sarcoma subtypes. Unsupervised double-hierarchical clustering placed DSRCT next to PC on branch 1. Other sarcoma subtypes clustered on branch 2. Distinct blocks within the heatmap indicate genes overexpressed (A) or underexpressed (B) in PC and DSRCT compared to other sarcoma subtypes. Within the branch 1, some genes were upregulated strictly in PC (C) compared to DSRCT (D). PC = prostate cancer (n = 12); OS = osteosarcoma (n = 47); CS = chondrosarcoma (n = 7); WDLPS = well-differentiated liposarcoma (n = 7), and DDLPS = dedifferentiated liposarcoma (n = 10).

AR-ASO (IONIS 560131; formerly AZD5312) significantly slowed DSRCT cell proliferation at two weeks (Fig. 4f) and reduced AR expression (Fig. 4g). However, the in vitro antiproliferative effect was four-fold more effective in the cells treated with the AR-targeted antisense blockade (Fig. 4e, f). Notably, this antineoplastic effect required concurrent administration with DHT (Supplementary Fig. S4). Altogether, these data indicate a vital role for DHT-stimulated AR expression in DSRCT cell proliferation and conclusively demonstrates a potent antineoplastic effect of AR antagonists.

**Preclinical efficacy of AR-based targeted therapy for the treatment of DSRCT.** Since only one DSRCT cell line exists, we extended our evaluation of the AR antagonists to the in vivo setting using the JN-DSRCT xenograft and available DSRCT PDXs. Immunocompromised NSG mice bearing JN-DSRCT xenograft tumors treated with enzalutamide or AR-ASO significantly reduced tumor burden and improved survival with the same efficacy, compared to placebo or control groups during the first two months of treatment (Fig. 5a, b). At 2 months, tumor growth began to accelerate in the enzalutamide-treated mice, whereas growth suppression continued in the mice treated with either 25 or 50 mg/kg of the AR-ASO (p < 0.0001; Fig. 5a). Compared to enzalutamide, the AR-targeted ASO (25 and 50 mg/ kg) demonstrated superior antineoplastic activity (Fig. 5a; p < 0.0001 or p = 0.006, respectively). The effects of AR-ASO and control ASOs were also assessed in NSG mice (5 mice/group)

bearing a DSRCT PDX (Fig. 5d–f). As expected, tumor growth and Kaplan–Meier curves revealed that tumors treated with AR-ASO have significantly reduced tumor burden and improved survival compared to control ASO group (p = 0.0097 and p < 0.0001, respectively).

Though both agents delayed tumor growth, AR-ASOs were more effective than enzalutamide in both preclinical models. Therefore, our pharmacodynamic analysis focused primarily on the effect of AR-ASO treatment. Proteomic profiling by RPPA (Fig. 6a), western blotting (Fig. 6b, c), immunofluorescence (Fig. 6d, e and Supplementary Fig. S5a–c), and immunohistochemistry (Fig. 6f–h and Supplementary Fig. S5d–f) validated the AR-ASO mediated knockdown of AR expression in the xenograft and PDX. To further characterize how the AR-ASO differed from enzalutamide mechanistically, we performed liquid chromatography-tandem mass spectrometry analysis of 38 collected preclinical animal specimens shown in Fig. 5. Consistent with prior literature in PC, loss of AR following AR-ASO treatment destabilized testosterone and reduced its intratumoral expression (Supplementary Fig. S6)[39,40]. As a negative control, the corticosterone levels were unchanged by AR blockade. In addition, since the antineoplastic action of enzalutamide works by preventing ligand-AR binding, reducing AR shuttling to the nucleus, and impairing AR DNA binding affinity—instead of reducing AR levels (Supplementary Fig. S7a–e)—enzalutamide treatment did not significantly lower intratumoral testosterone.

To gain a preliminary understanding of the short-term pharmacodynamic effects of AR suppression, a group of JN-DSRCT xenografts and DSRCT PDXs was collected 10 days

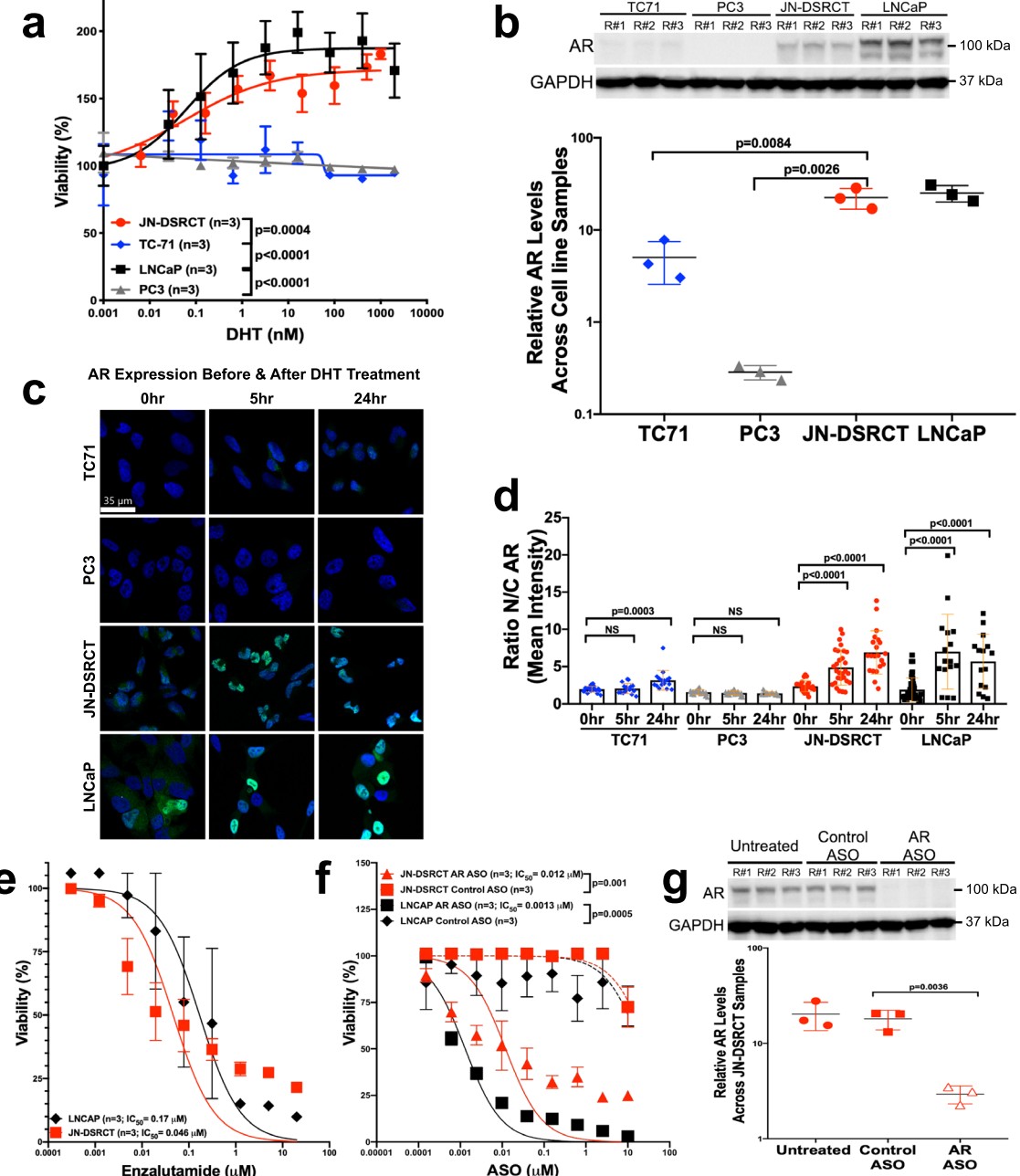

**Fig. 4 In vitro stimulation and inhibition of DSRCT proliferation via AR. a** JN-DSRCT, TC71, LNCaP, and PC3 cell proliferation assays after treating them with an AR agonist hormone, dihydrotestosterone (DHT) in a dose-dependent manner. *n* is the number of experimental replicates. **b** Profiling JN-DSRCT, TC71, and LNCaP cells for their AR expression by western blotting and histogram presentation of relative AR levels across each cell line. **c** Profiling of JN-DSRCT cells for AR protein expression (green) by immunofluorescence analysis with DAPI-labeled nuclei (blue), **d** Quantitative scatter plot representation of the ratio Nuclear/Cytoplasmic AR mean intensity reported within a single cell at 0, 5 and 24 h of DHT post-treatment. **e** JN-DSRCT cells are relatively less sensitive to enzalutamide than (**f**) AR antisense oligonucleotides treatment, as shown by the in vitro WST1-Proliferation cell-based assay. **g** Western blot analysis of AR expression in JN-DSRCT cells untreated or after Control-ASO and AR-ASO treatments. Histogram presentation of relative AR levels across each cell line after GAPDH normalization. Data points in **a**, **e**, and **f** represent mean ± SEM using three experimental replicates for each cell line. Data in **b** (bottom panel), **d** and **g** (bottom panel) represent mean ± standard deviation. *P* values calculated by unpaired two-tailed *t*-test.

into their AR-ASO treatment (Fig. 6a, AR-ASO PD) for analysis by RPPA to assess early compensatory pharmacodynamic changes. pS6, Akt, estrogen receptor (ER), PD-1L, pAKT, and other proteins (Fig. 6a and Supplementary Fig. S7f–g) were upregulated. Others have reported that the PI3K-AKT pathway has pleiotropic effects on survival, proliferation, metabolism, and growth pathways of several malignancies[41], and its blockade has long been of interest in managing PC, where a compensatory

increase in AKT signaling can occur following AR inhibition[42]. Notably, the same AR-ASO (AZD5312) used in our preclinical experiments was well tolerated when administered to PC patients (NCT03300505). Therefore, one could theoretically investigate this AR-ASO drug candidate in DSRCT-specific phase 2 trials without delay. Given the limited nature of our preclinical studies, future studies with enzalutamide are also of interest.

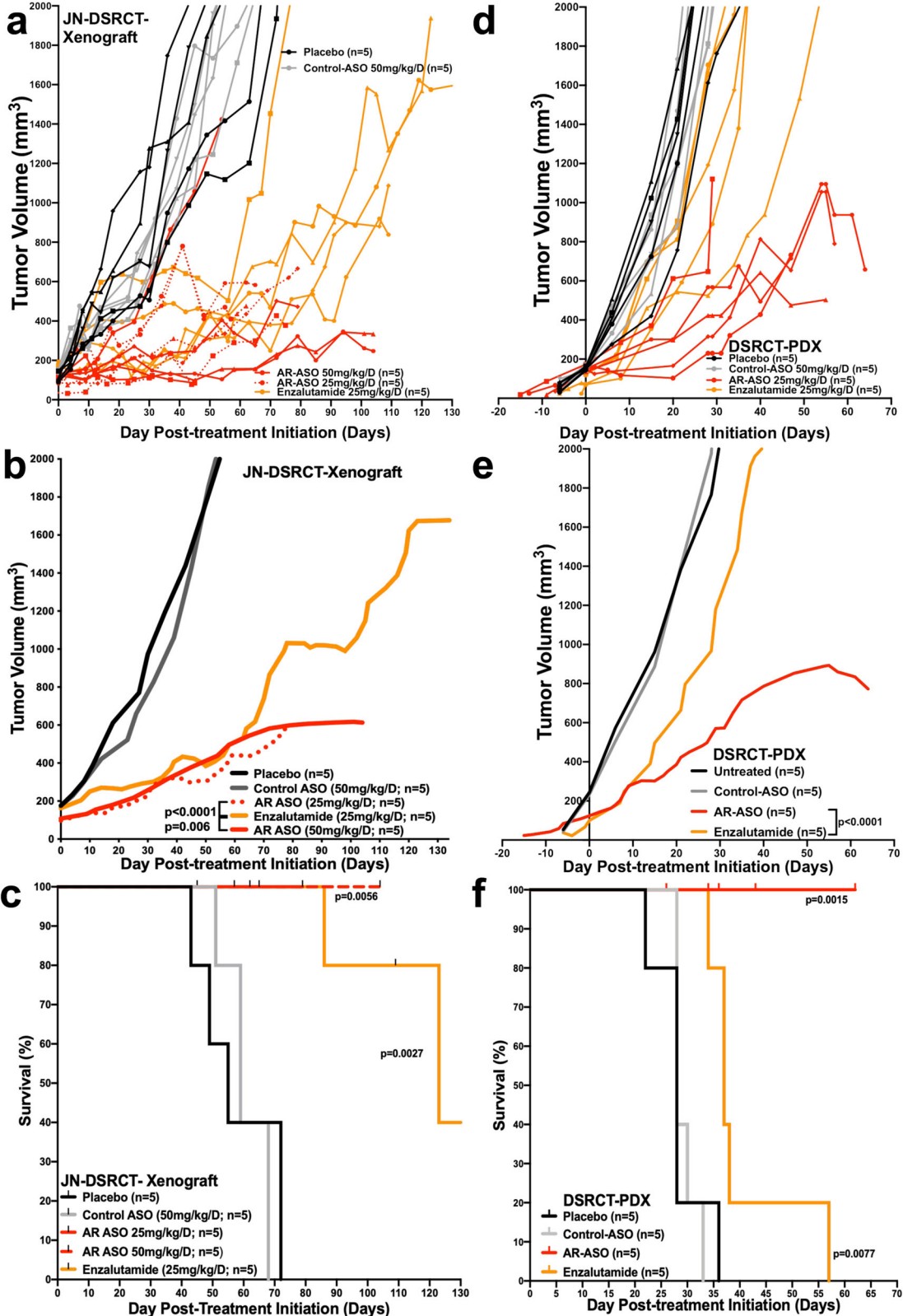

**Fig. 5 Preclinical efficacy of AR antisense-based therapy for the treatment of DSRCT. a–c** Therapeutic effect of AR blockade in JN-DSRCT xenografts done in three replicates. Tumor-bearing mice volumes and survival were reported after treatment with enzalutamide (25 mg/kg, orange), AR-ASO (25 mg/kg, regular red; 50 mg/kg, dotted red), control ASO (gray), or placebo (black). The top panel (**a**) shows the individual data for each mouse; the middle panel (**b**) shows a smoothed grouped median of relative tumor volumes; the lower panel (**c**) shows the survival Kaplan–Meier curves of each treated group of mice. **d–f** Similar data is shown for a DSRCT PDX treated with the same agents. **d** Individual PDX data, **e** smoothed PDX data, and **f** Kaplan–Meier curves for the DSRCT PDX. *P* values reported for the smoothed tumor growth curves (**b** and **e**) were calculated by a two-tailed unpaired *t*-test. Kaplan–Meier *P* values were calculated by the log-rank (Mantel–Cox) test. *n* is the number of mice treated in each treatment group.

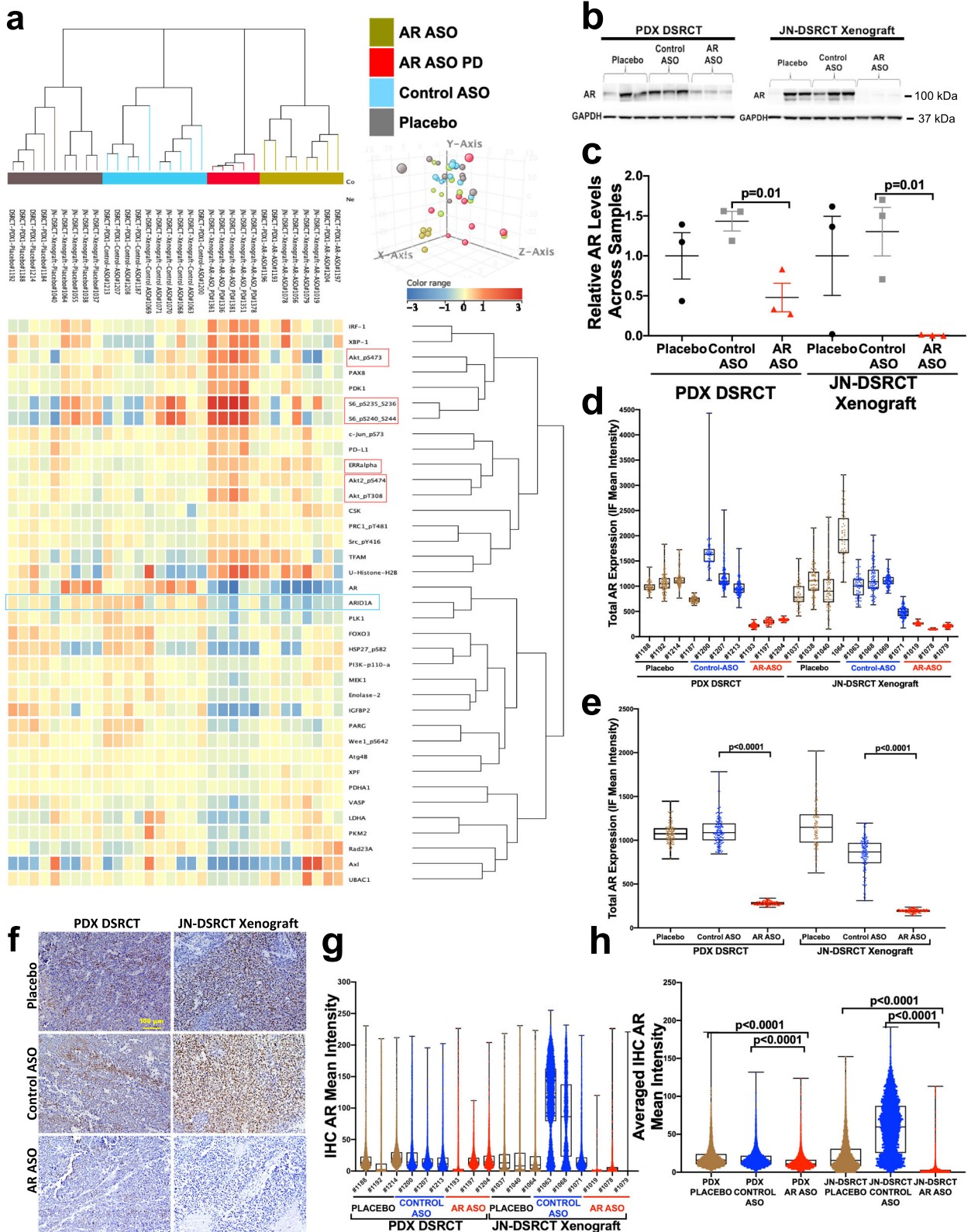

**AR directly regulates important oncogenic regulators in DSRCT**. To model the AR transcriptional program in a human JN-DSRCT cell line, we determined the genome-wide AR binding profiles using ChIP-Seq experiments in unstimulated or DHT-stimulated JN-DSRCT cells treated with control ASO or AR-ASO. As expected, DHT treatment enhanced AR binding to the chromatin as assessed by the average intensity plot on all significant

peaks ($p < 1e^{-7}$) and heatmap (Fig. 7a). DHT stimulation led to ~4000 new peaks that were suppressed by treatment with AR-ASO (Fig. 7b and Supplemental Data 1). These binding sites were enriched at known AR response elements (AREs) and in sites for *FOXA1*, a transcription factor known to open compacted DNA and cooperate with AR in PC[43] (Fig. 7c and Supplemental Data 1). Consistent with DSRCT's pathogenesis, we also noted

**Fig. 6 Proteomic evaluation of AR expression in JN-DSRCT and PDX tumors after AR-based antisense therapy. a** The principal components analysis plot and reverse-phase protein lysate array (RPPA) evaluations of JN-DSRCT and PDX tumors after therapies, separated the 32 samples into four groups and identified 37 proteins statistically significantly associated with the treatment at a false discovery rate (FDR) of 0.05. **b** Immunoblotting evaluation of JN-DSRCT xenograft and PDX-DSRCT tumors after AR-ASO treatment. **c** AR normalization relative to GAPDH within the preclinical tumor samples. AR biomarker was significantly reduced in mice treated with AR-ASO compared to the control ASO group ($p = 0.01$). **d** Representative AR immunofluorescence confocal microscopy quantification of the preclinical JN-DSRCT and PDX tumor samples, within the single cell or **e** the averaged treated samples (placebo, control ASO, and AR-ASO). **f** Immunohistochemical evaluation images of preclinical JN-DSRCT and PDX1 tumor samples. IHC stains for AR in primary tumors of JN-DSRCT and PDX-DSRCT mice after treatment with AR-ASO, control ASO, and placebo. 100 μm scale bars are shown. **g** Representative IHC AR mean intensity quantification of the preclinical JN-DSRCT and PDX tumor samples, within the single cell or **h** the averaged treated samples (placebo, control ASO, and AR-ASO). All tumors analyzed by RPPA were collected at tumor progression or the experiment's conclusion, except for the AR-ASO PD group, which was collected 10 days after initiating therapy to enable pharmacodynamic analysis. Data in **c**, **d**, **e**, **g**, and **h** represent mean ± standard deviation. *P* values calculated by unpaired two-tailed *t*-test.

enrichment of *WT1* binding motifs within AR binding peaks (Fig. 7c) suggesting potential interactions between AR, FOXA1, and WT1 in JN-DSRCT cells. To further characterize the genes adjacent to AR binding site peaks, we performed a pathway analysis using 700 genes that are direct targets of AR. Upregulated pathways included the TNFα pathway, Hippo signaling, and pluripotency regulators (Fig. 7d and Supplemental Data 1), and key genes included *WT1*, *CTNNB1*, *SOX2*, *GLI2*, *FOXF1*, and *GATA6* (Fig. 7e, Supplementary Fig. S8d, and Supplemental Data 1).

After evaluating the effects of androgen stimulation and withdrawal in JN-DSRCT cells, we next compared DSRCT to data from PC cells. Significant overlap existed at sites for AR binding at AREs (Supplementary Fig. S8a), FOXA1 motifs (Supplementary Fig. S8b), and sites that regulate key cancer pathways, including WNT, TGFβ, PI3K, MAPK, Hippo signaling, TNFα, and epithelial-to-mesenchymal transformation (Supplementary Fig. S8c). To further evaluate the AR regulatory function in DSRCT tumor mouse models, we performed ChIP-seq on DSRCT-xenograft and PDX samples. Consistent with the cell line data, we observed suppressed AR binding to the chromatin by the treatment with AR-ASO (Supplementary Fig. S9a, b). Similarly, pathway analysis of the top 5000 lost AR binding sites targeted genes showed enrichment of MAPK pathway, Hippo signaling, Wnt signaling, and pluripotency regulators (Supplementary Fig. S9c, d). We also noted enrichment of AR and FOX family binding motifs within AR binding peaks in both DSRCT-xenograft and PDX samples (Supplementary Fig. S9e). Genes adjacent to AR binding site peaks also showed high overlap with DSRCT-specific genes in both models (Supplementary Fig. S9f). Key genes from cell line data (Fig. 7e) also showed AR signal reduction after AR-ASO treatment (Supplementary Fig. S9g).

**AR-dependent enhancer reprogramming activates oncogenic pathways in DSRCT.** Several studies have shown that AR establishes a pro-tumorigenic transcriptome by reprogramming the active enhancer landscape (assessed by H3K27ac profiles) in PC progression[44]. Therefore, we asked if AR plays similar roles in DSRCT by examining genome-wide profiles for H3K27ac marks in unstimulated or DHT-stimulated JN-DSRCT cells treated with control ASO or AR-ASO. We noted that unstimulated cells treated with AR-ASO showed a higher intensity and a higher number of H3K27ac peaks compared to control ASO-treated cells (Fig. 8a, b, Supplementary Fig. S10a, and Supplemental Data 2). Similarly, AR-ASO treatment in DHT-treated cells also increased the active enhancer peaks compared to control ASO treatment (Fig. 8a, b, Supplementary Fig. S10a, and Supplemental Data 2). This observation is contrary to those in PCs where active enhancer peaks are positively associated with higher AR activity[44]. It has been previously shown that AR recruits the MLL complex and CBP/p300, which is responsible for active enhancer

marking in PC[45]. To identify which enhancers were likely derived by AR binding and potential recruitment of enhancer-marking proteins, we overlapped the AR and H3K27ac peaks in DHT-treated cells (Supplementary Fig. S10b and Supplemental Data 2). We then intersected these AR-targeted enhancer peaks with highly expressed genes in DSRCT (Fig. 8c and Supplemental Data 2) (FC > 1.5, adjusted *p* value < 0.05 in comparison to other sarcoma subtypes). There, we identified WNT signaling and cell-adhesion as major drivers that are regulated at the chromatin level by AR-dependent active enhancer programs (Fig. 8d). The genes with direct AR binding and enhancer gains included important oncogenes such as *AXIN2* and *CDK6* (Fig. 8e). In addition, we investigated alterations in super-enhancer (SE) regions that harbor a high density of TF binding motifs[46–48]. SEs in control ASO-treated cells marked important oncogenes such as *AKT3* and *GRHL2*, whereas SEs in AR-ASO-treated cells marked tumor suppressor genes such as *RUNX1* and *CUX1* (Supplementary Fig. S10c, d and Supplemental Data 3), that potentially regulate the AR-driven transcriptome. Overall, our results suggest that AR activation reprograms typical enhancers and SE to regulate key oncogenic signaling pathways in DSRCT.

Interestingly, in preclinical tumor samples, we also observed similar enhancer reprogramming. AR-ASO treatment of the DSRCT-xenograft significantly increased the active enhancer and promoter binding sites compared to control ASO treatment, whereas PDX samples showed a moderate increase (Supplementary Fig. S11a–d). We also observed AR-dependent active enhancers regulating PI3K-AKT-mTOR, WNT signaling, cell-adhesion pathways (Supplementary Fig. S11e, f), and key oncogenes (Fig. 8e), with direct AR binding and enhancer gains in both DSRCT-xenograft and PDX models (Supplementary Fig. S11g).

**Discussion**

Ever since Ladanyi and Gerald discovered the EWSR1-WT1 chromosomal translocation[8], DSRCT has been treated with the same chemotherapy regimens used for ES. The recent exceptions include ES-specific agents like TK-216 that target c-terminus ETS genes (e.g., FLI1 or ERG), or pazopanib, which demonstrates preferential activity in DSRCT and other soft-tissue sarcomas[49]. Phase II studies testing neoantigen targeted monoclonal antibodies, for example, 8H9 in DSRCT, are also directed at unique sarcoma subtypes[50].

As three-quarters of all DSRCT patients typically succumb to their malignancy within 5-years, our RPPA study intended to define new molecular targets for DSRCT and expand our therapeutic arsenal of biologically targeted therapies that engage them (Fig. 1). Surprisingly, of 151 proteins assessed in the RPPA, SYK and AR were the most differentially expressed. The SYK protein —not previously reported in DSRCT—is a non-receptor tyrosine kinase (also known as spleen tyrosine kinase) commonly found in

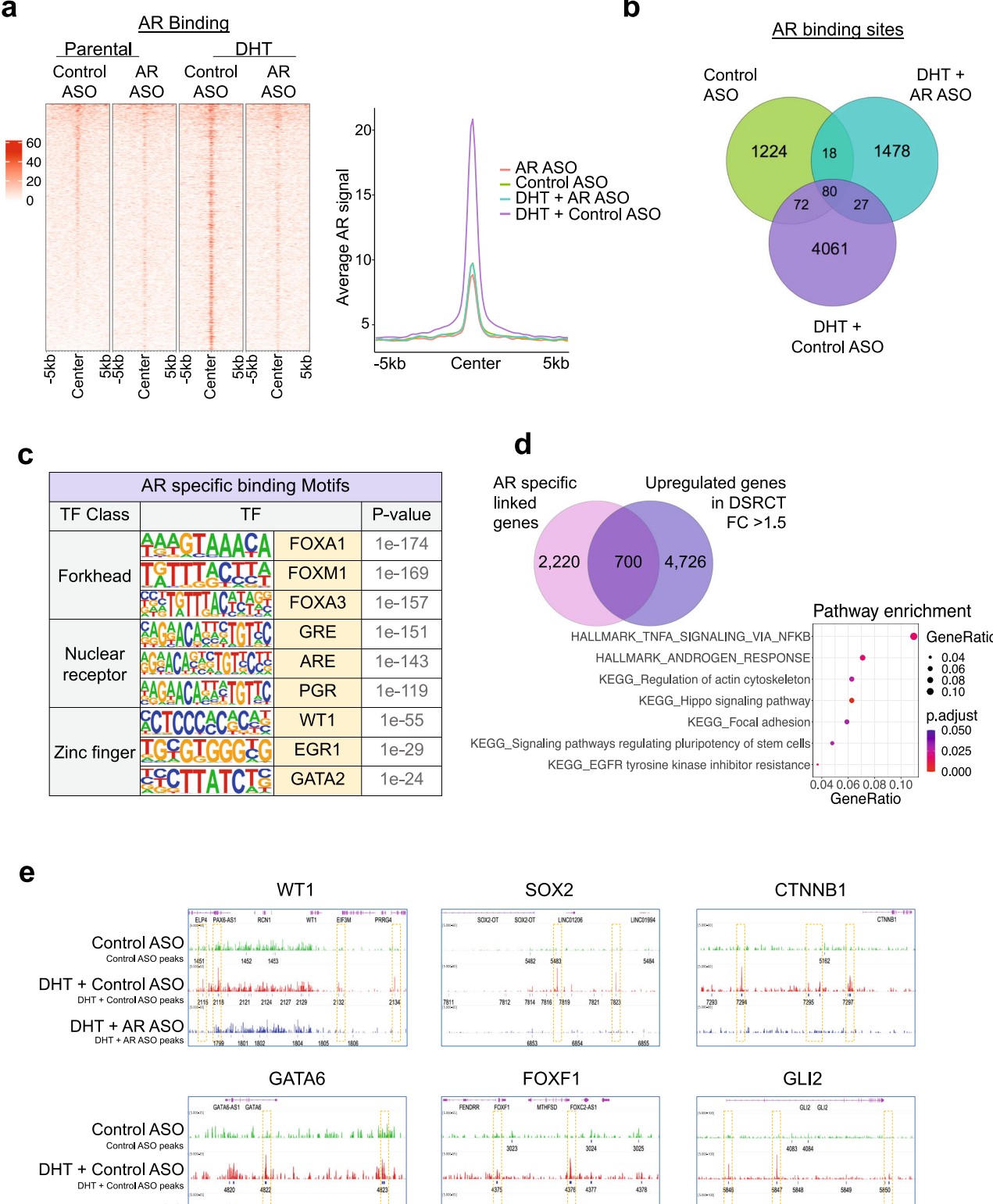

**Fig. 7 AR binding in JN-DSRCT cells. a** Heatmaps (left panels) and average intensity curves (right panels) of ChIP-seq reads (RPKM; reads per kilobase of transcript per million mapped reads) for AR binding regions. AR binding sites are shown in a 10-kb window (centered on the middle of the binding site) in Control ASO, AR-ASO, DHT + Control ASO, and DHT + AR-ASO samples. **b** Venn diagram showing the overlap of all AR peaks between Control ASO, DHT + Control ASO, and DHT + AR-ASO samples to identify the AR-unique or shared binding sites. **c** List of enriched transcription factor (TF) motifs in AR-specific binding sites. Motifs are identified using HOMER (Binomial test). **d** Dot plot showing significantly enriched pathways for AR-specific binding sites. Dot size represents gene ratio, and colors represent adjusted *p* values (Fisher's exact test). **e** IGV images showing enrichment of AR peaks around *WT1, SOX2, CTNNB1, GATA6, FOXF1,* and *GLI2* genes using aggregate ChIP-seq profiles of Control ASO, DHT + Control ASO, and DHT + AR-ASO samples.

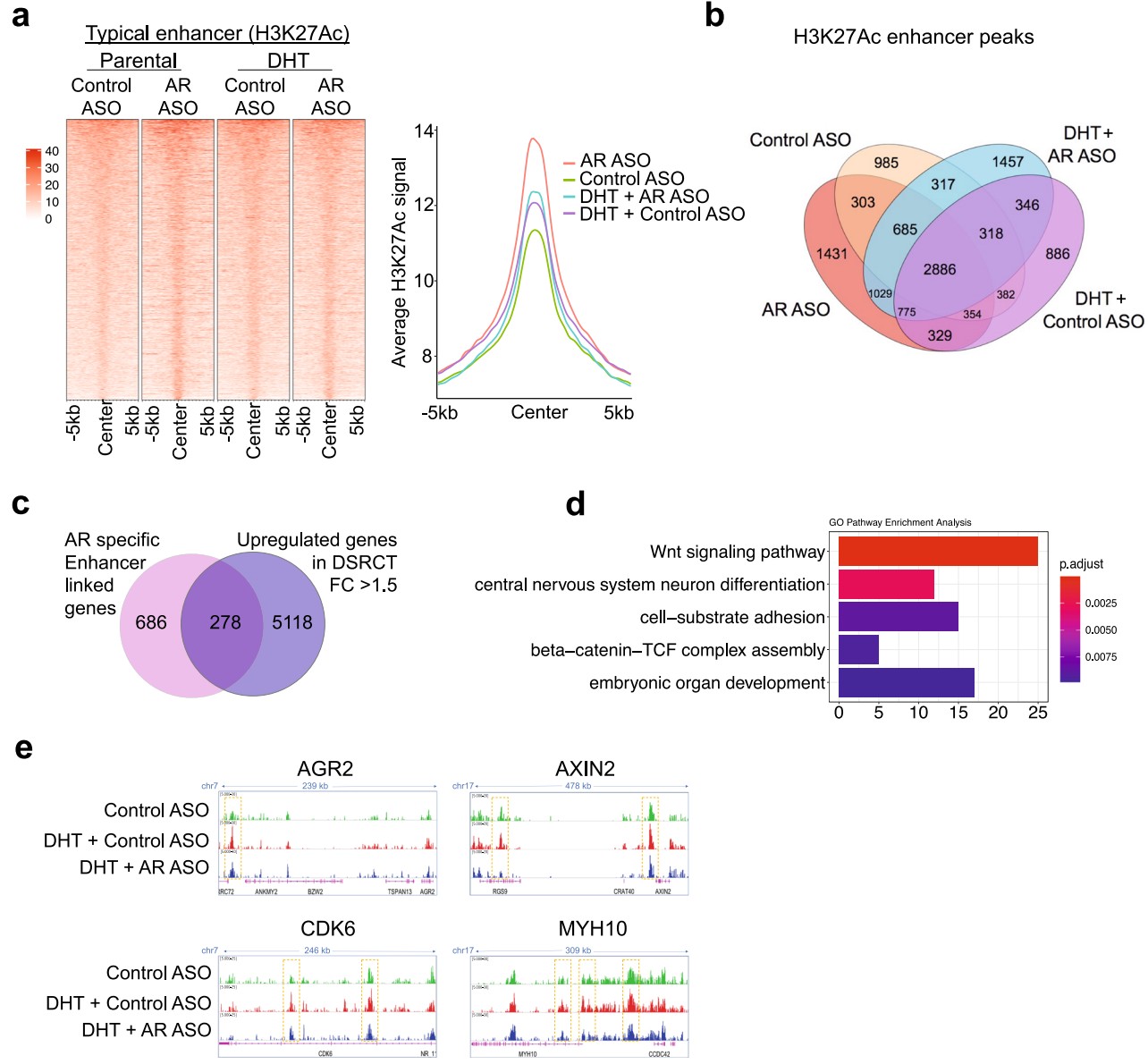

**Fig. 8 Enhancer reprogramming by AR in JN-DSRCT cells. a** Heatmaps (left panels) and average intensity curves (right panels) of ChIP-seq reads (RPKM; reads per kilobase of transcript per million mapped reads) for typical enhancer regions. Enhancer regions are shown in a 10-kb window (centered on the middle of the binding site) in Control ASO, AR-ASO, DHT + Control ASO, and DHT + AR-ASO samples. **b** Venn diagram showing the overlap of all enhancer peaks between Control ASO, AR-ASO, DHT + Control ASO, and DHT + AR-ASO samples to identify the AR-unique or shared enhancer reprogramming. **c** Venn diagram showing the overlap of annotated genes for AR-specific gained enhancer peaks and upregulated gene list for DSRCT tumors vs. other sarcoma tumors to identify the AR-unique enhancer reprogramming associated transcription upregulation. **d** Bar plot showing significantly enriched pathways for AR-specific enhancer reprogramming associated transcription upregulation. Bar length represents gene numbers, and colors represent adjusted *p* values (Fisher's exact test). **e** IGV images showing enrichment of H3K27Ac peaks around *AGRE2*, *AXIN2*, *CDK6*, and *MYH10* genes using aggregate ChIP-seq profiles of Control ASO, DHT + Control ASO, and DHT + AR-ASO samples.

hematological tissues. Its constitutive activation has been shown to induce malignant transformation of B-cells to lymphomas or leukemias. As such, the oral SYK inhibitors cerdulatinib (Portola Pharmaceutical) and entospletinib (Gilead Sciences) are under active clinical investigation for the treatment of certain lymphomas, chronic lymphocytic leukemia, and acute myeloid leukemia (NCT01994382 and NCT02457598). An orally active SYK inhibitor, fostamatinib, has already received FDA approval as a treatment for immune thrombocytopenia and continues to be investigated as an experimental therapy for hematological malignancies (NCT00446095). Though tantalizing to consider that SYK hyperactivation plays an oncogenic role in DSRCT, we

have not yet had the opportunity to evaluate these relatively new drugs within our preclinical DSRCT models.

In contrast to SYK, numerous FDA-approved and experimental AR antagonists were available for immediate preclinical evaluation, and potentially available to patients via compassionate access or early-phase clinical trials. Though our RPPA data and 9:1 male-to-female ratio hinted that DSRCT is an AR-driven malignancy, to prove this explicitly we proposed several criteria, akin to Koch's postulates: (a) tumors must adequately express AR, (b) DHT must stimulate DSRCT cell proliferation, and (c) AR antagonists should curtail the tumor-promoting effects of androgen stimulation. The inclusion of mechanistic studies,

including those directly tying AR to AREs, lends further credibility that DSRCT is a second AR-driven malignancy.

To date, the first criterion—requiring AR expression—has been reported by two prior teams that recognized the striking predilection of DSRCT for young males[15,51]. As discussed briefly in the Introduction, Fine et al. evaluated protein expression of AR, c-Kit, EGFR, and other proteins by WB and IHC, scored using a 5-point scale that ranged between 0 (no staining) to 4+ (highly positive)[15]. Ten of 27 (37%) DSRCT patients in their case series scored 2+ or higher, but we highlight that 15 demonstrated no AR expression (Fig. 2), which suggests prospective studies may wish to stratify for response by AR status to determine whether AR expression correlates with therapeutic efficacy. A more recent study published in 2018 by Bulbul et al. at U.C.S.D., used IHC and next-generation sequencing on tumors from 35 DSRCT patients (86% who were males); 59% were AR-positive using a dichotomous cut-off that required ≥1+ staining in ≥10% of the cells[51]. In the present study, we report the most extensive series of DSRCT patients to have undergone protein and transcriptomic profiling. Though enriched in oncoproteins, our RPPA array ranked AR as the most differentially expressed protein compared to ES, its closest molecular sarcoma subtype (Fig. 1). Our subsequent confirmation of the RPPA results by WB, and later semi-quantitative analysis by IHC, agrees with earlier reports and appears to substantiate AR as a bona fide target in DSRCT.

Meeting the second of Koch's postulates, a 72-h cell proliferation assay demonstrated a significant increase in JN-DSRCT cell proliferation following exposure to physiological levels of DHT (Fig. 4a), though lower than LNCaP PC cells. As one would expect in androgen-sensitive cells, DHT also promoted the nuclear shuttling of AR into the nucleus where it would function as a transcriptional regulator of its target genes (Fig. 4c, d). As our results rely upon data from a single cell line (the only one in existence at the time), we remain vigilant to avoid over-interpreting them. Nevertheless, our results echo similar findings by Fine et al., where they reported a DHT-induced two-fold increase in cell proliferation in a transient DSRCT cell line obtained from ascitic fluid[18].

Fulfilling the third requirement that defines an AR-driven malignancy, our team again bolsters the work by Fine et al., which had taken a prescient step more than a decade ago to evaluate CAB—in that case, using Lupron and bicalutamide in six DSRCT patients that were AR-positive (3+ or 4+ by IHC)[18]. Interestingly, in their limited pilot trial, non-castrate level baseline testosterone levels were associated with modest responses lasting 3–4 months. Admittedly, having tested several DSRCT patients with the same drug combination between 2006 and 2015, well before the advent of modern-day androgen deprivation therapies such as abiraterone and enzalutamide, our team observed limited clinical benefits lasting <3 months. Our renewed enthusiasm for AR targeting in DSRCT stemmed from the RPPA expression results, accompanied by the in vitro DHT stimulation studies and in vivo data using enzalutamide and the AR-ASO (Fig. 5).

In preparation for early-phase clinical trials now in development, our work takes the first step to advance our mechanistic understanding of AR signaling in DSRCT. As one of several steroid and nuclear hormone superfamily receptors that include the glucocorticoid receptor, mineralocorticoid receptor, progesterone and estrogen receptors, and the vitamin D receptor, AR retains a conserved 66-amino acid DNA-binding domain able to join two (5'-AGAACA-3') hexameric half-sites arranged as an inverted palindrome spaced 3-b.p. apart (IR3). Due to differences in local steroid metabolism, ligand abundance, chromatin accessibility, and cofactor occupancy, the DNA binding pattern of AR varies significantly in PC compared to other tissues[52]. Interestingly, among the pioneer factors that govern the lineage-specific

binding of AR to specific genomic loci in PC[53–55], and that control AR-mediated transcriptional regulation of prostate genes (such as PSA)[56], FOXA1 was the most enriched MOTIF in JN-DSRCT cells (Fig. 7c). Shared activation of the androgen signaling cascade in DSRCT and PC may explain the close transcriptomic clustering observed in Fig. 2e. Despite their similarities, ChIP-seq also identified notable differences in AR's epigenetic regulation at enhancer (Fig. 8) and SE (Supplementary Fig. S9c) binding sites.

Though the subject of future research, we suspect the heterotypic loss of WT1 or aberrant EWS-WT1 FP may recruit a specific set of chromatin modifiers at binding sites that differ from PC. Others have performed ChIP-seq in DSRCT patient specimens using WT1-specific antibodies, but the Santa Cruz antibody used in that publication[57] has been discontinued. Lacking suitable ChIP-seq validated WT1-specific antibodies ATAC-seq might be used before and after WT1 RNA silencing, though the interpretation of that experiment would not be as straightforward given the absence of selective antagonism of WT1 or EWS-WT1.

Interestingly, as occurs in castration-resistant PC[11,58,59], our pharmacodynamic studies revealed an inverse relationship between AR and the Akt/PI3K/mTOR pathway. With numerous inhibitors of PI3K and mTOR already FDA approved, an obvious next step would be to investigate whether co-targeting AR and either PI3K or mTOR results in synergistic anti-cancer activity. Though not explored in the present study, the ER was also highly expressed following AR-ASO treatment. Given the shared binding DNA motifs that ER, AR, and other steroid hormone receptors have in common, this observation suggests that ER-targeted drugs might prove useful for patients with castrate-resistant DSRCT and, plausibly, the small minority of women that acquire this rare cancer type. Of course, further research is required to determine how AR and ER pathway switching affects tumor growth and survival, both in DSRCT and other hormonally-driven malignancies[60].

Collectively, though morphologically and phenotypically distinct from PC, our data suggest that DSRCT is a second androgen-stimulated malignancy (third, if one considers the AR-positive molecular subset of triple-negative breast cancer). Shared dependence upon AR for tumor growth and survival provides an exciting opportunity to study AR signaling in a different cancer type and within a younger DSRCT-stricken patient population. Preclinical data using enzalutamide and AR-ASO raises the tantalizing possibility that AR-targeted drugs used for PC may also find utility to combat DSRCT.

## Methods

**Tumor specimens.** In compliance with all relevant ethical guidelines, patients provided written informed consent to collect and use their tumor specimens for research purposes using lab protocols LAB08-0151 or LAB04-0890, which are approved by MDACC's Institutional Review Board. The charts and electronic medical records of patients with a confirmed diagnosis of DSRCT were included for analysis and archived at the MDACC biospecimen bank or the collaborator PIs laboratories. We identified 60 DSRCT patients treated at MDACC from 1990 to 2019 to generate a TMA. Also, we collected 16 DSRCT and 6 ES fresh frozen tumors, all of them were profiled by RPPA. Specialist pathologists used clinical information, immunohistochemistry, and cytogenic analysis for the EWSR1-WT1 or EWSR1-FLI1 fusions to confirm the DSRCT or ES diagnoses.

**RPPA and western blot analyses.** The available snap-frozen DSRCT ($n = 16$) and ES ($n = 6$) specimens collected during a core-needle biopsy or surgical debulking procedures using clinical protocols approved by MDACC's Institutional Review Board and specimens of normal-appearing mesenteric tissue adjacent to DSRCT obtained at the time of surgical debulking ($n = 8$) were used for the proteomic analysis (Supplementary Table 1: Aggregate demographic information of DSRCT and ES patients). Lysates were created, protein concentrations were determined, and individual protein expression was measured using a well-validated RPPA and WB technologies as previously described[61–63]. AR protein detection was performed using the CST antibody (#5153). Additional details about RPPA and WB analyses and normalized data are provided in the Supplementary Methods and Supplemental Data 4.

**RNA sequencing, gene expression analysis, and fusion detection**. Total RNA from primary tumor samples was extracted and libraries were made from cDNA using the NuGEN Ovation Ultralow Library System V2 (San Carlos, CA). RNA sequencing reads of the samples were mapped to the hg19 reference genome using the STAR aligner[64]. For calculation of gene expression, each gene's raw count data were first obtained using HTSeq[65], and are normalized by scaling the library size using calcNormFactors in the edgeR package[66]. Then, Voom transformation was applied to normalized counts and a linear model fit to the data for differential expression analysis using the Limma package[67]. Pathway analyses of differentially expressed genes between two sample clusters were performed using Gene Set Enrichment Analysis[68]. Fusion transcripts were detected from RNA-seq data using MapSplice[69].

**TMA preparation and immunohistochemistry analyses**. A TMA was constructed from archival surgical pathology materials comprising 60 formalin-fixed, paraffin-embedded (FFPE) tissues from 60 DSRCT patients. Areas of the viable tumor were selected by pathologist review of whole slide H&E-stained sections. Selected areas were punched and transferred, in duplicate, to a recipient block using an ATA-100 Advanced Tissue Arrayer (Chemicon International). All human specimens were utilized under an Institutional Review Board-approved research protocol (LAB04-0890) allowing for the retrospective sampling and analysis of existing archival materials collected during standard patient care. Immunohistochemical studies were performed using an autostainer (Bond-Max; Leica Microsystems, Buffalo Grove, IL, USA) with anti-AR (1:30; clone AR441, Dako#M3562) antibody. Additional details about TMA slide preparation and IHC analyses are provided in the Supplementary Methods.

**WST1 cell proliferation assays**. The JN-DSRCT, LNCaP, and TC71 cells were tested for their proliferation capacity in vitro using a colorimetric assay in 96-Well plates with WST1 reagent (Roche). The cells were seeded at 3000 cells/well in triplicates with 10% FBS DMEM (JN-DSRCT) or RPMI (TC71 and LNCaP) complete media. Additional details about WST1 cell proliferation assays are provided in the Supplementary Methods. JN-DSRCT and TC71 cell lines were supplied by JL. LNCaP and PC3 were obtained through MDACC's cell line repository.

**Immunostaining of JN-DSRCT cells and xenograft animal tumors**. JN-DSRCT cell line exhibiting a pathognomonic t(11;22)(p13;q12) translocation was generously provided from Dr M. Kikuchi's laboratory (Fukuoka University, Fukuoka, Japan). In addition, PC3, LnCaP, and TC71 cell lines are provided by the MDA cell lines core facility. All available cell lines in Dr Ludwig's lab are registered within the MDA characterized cell line core (CCLC). Each cell line identity is validated twice per year in MDA CCLC using short-tandem repeat fingerprinting with an AmpFLSTR Identifier kit. Furthermore, according to the manufacturer's protocol, all our cell lines are tested once per year for mycoplasma contamination using the MycoAlert Detection Kit (Lonza Group Ltd.). In addition, cell lines are sent for third-party mycoplasma testing using a sensitive PCR testing approach any time a collection of cells are cryopreserved.

Monolayer JN-DSRCT cell culture in 8 chamber slides was fixed for 10 min at room temperature with 4% paraformaldehyde in phosphate-buffered saline (PBS). The primary JN-DSRCT xenograft and PDX tumors were harvested, fixed in 10% formalin, embedded in paraffin (FFPE), and then sliced in 5-μm sections before processing them for antigen retrieval using 0.1 M citrate buffer for 20 min and in a vegetable steamer. Altogether, monolayer and primary tumor slides were permeabilized and blocked with superblock buffer (Thermo Fisher Scientific, #37535) for 1 h at room temperature. Slides were then incubated consecutively with primary antibodies to AR (Cell Signaling Technologies, #5153), (overnight at 4 °C) and Alexa Fluor 488-labeled Goat-anti-Rabbit (Thermo Fisher Scientific, #A11037) (for 1 h at room temperature). The nuclei were visualized using Hoechst (Thermo Fisher Scientific, #H357), and the immunofluorescence was acquired after subtracting the background intensities using the Nikon A1-Rsi confocal microscope (Nikon). Fluorescent detection of proteins in the nuclei and cytosolic regions was quantified using the Imaris software (Bitplane) and its Cell module that use validated algorithms to define the segmentation by permitting the recognition of selected protein fluorescence in both nuclear and cytosolic regions.

**Generation of DSRCT xenograft/PDX mouse models and drug evaluation**. All experiments were conducted per protocols and conditions approved by the University of Texas MDACC (Houston, TX) Institutional Animal Care and Use Committee (eACUF Protocols #00000712-RN03). Male NOD (SCID)-IL-2Rg[null] mice 6 weeks in age (The Jackson Laboratory; Farmington, CT) were subcutaneously injected with JN-DSRCT cells ($5 \times 10^6$ cells/animal) or received PDX explants (2 mm) to generate DSRCT xenograft and PDX mouse models. The histologic and genetic analyses of DSRCT patients and PDX tumors are available in Supplementary Fig. S12. All mice were maintained under barrier conditions and treated using protocols approved by The University of Texas MDACC's Institutional Animal Care and Use Committee. Once their tumors reached a volume of 150 mm³, 5 mice per group received enzalutamide (25 mg/kg IP daily, 5 times per week), or AR-ASOs (25 or 50 mg/kg subcutaneously daily, 5 times per week), or control ASOs (50 mg/kg subcutaneously daily, 5 times per week), or a placebo

control (sterile vehicle buffer). Tumor volumes were measured using digital calipers at study initiation and 2–5 times per week after that for up to 85 days, or until their tumors reached 1500 mm³, whichever came first. Per institutional requirements, tumor size never exceeded 2 cm in maximal linear dimension. A Kaplan–Meier analysis was performed to assess drug efficacy. Statistical analyses between the control and treated groups or between different treated groups were performed with the log-rank (Mantel–Cox) test using GraphPad Prism 8.0.

**Mass spectroscopy-based determination of intratumoral hormone levels**. Testosterone and corticosterone quantification was determined using Agilent's Infinity II UHPLC in line with a 6495 triple quadruple mass spectrometer and MassHunter workstation software (8.0.8.23.5). Briefly, DSRCT xenograft and PDX samples were homogenized using water containing internal standard (Cerilliant, T070) extracted with tert-butyl methyl ether (Sigma 34875), dried under nitrogen, and derivatized using hydroxylamine hydrochloride (Sigma 431362). The recovered ketoxime steroids were reconstituted in methanol/water (1:1 v/v) and injected into the Infinity II UHPLC. Ketoxime steroids were separated using a Chromolith reverse-phase column (RP-18 endcapped 100–2 mm, Sigma 152006) and introduced into a JetStream source (Agilent) for triple quadrupole analysis. Data were analyzed and quantified using MassHunter software (Agilent)[39,40].

**ChIP-seq assays**. Chromatin immunoprecipitation was performed as described earlier[70] with optimized shearing conditions and minor modifications for JN-DSRCT cells. The antibodies used were: H3K27ac (Abcam ab4729) and AR (CST#5153). Briefly, 3 million cells per sample were cross-linked using 1% formaldehyde for 10 min at 37 °C. After quenching with 150 mM glycine for 5 min at 37 °C, cells were washed twice with ice-cold PBS and frozen at −80 °C for further processing. Later, cells were thawed on ice and lysed with ChIP harvest buffer (12 mM Tris-HCl, 0.1 × PBS, 6 mM EDTA, 0.5% sodium dodecyl sulfate [SDS]) for 30 min on ice. Lysed cells were sonicated with Bioruptor (Diagenode) to obtain chromatin fragments. Antibody-dynabead mixtures were incubated for 1 h at 4 °C and cellular extracts were then incubated overnight with these mixtures. After overnight incubation, immune complexes were washed five times with RIPA buffer, twice with RIPA-500 (RIPA with 500 mM NaCl) and twice with LiCl wash buffer (10 mM Tris-HCl pH8.0, 1 mM EDTA pH8.0, 250 mM LiCl, 0.5% NP-40, 0.1% deoxycholate). For reverse-crosslinking and elution, immune complexes were incubated overnight at 65 °C in elution buffer (10 mM Tris-HCl pH8.0, 5 mM EDTA, 300 mM NaCl, 0.5% SDS). Eluted DNA was then treated with proteinase K (20 mg/ml) and RNase A and DNA clean-up was done using SPRI beads (Beckman-Coulter). ChIP libraries were amplified and barcoded with use of the NEB-Next® Ultra™ II DNA library preparation kit (New England Biolabs). After library amplification, DNA fragments were size-selected (200–500 bp) using AMPure XP beads (Beckman Coulter) and assessed using high sensitivity D1000 screen tape on the Bioanalyzer (Agilent Technologies). Libraries were multiplexed together and sequenced in HiSeq2000 (Illumina).

**RPPA and western blot analyses**. Raw log2 intensity values were normalized for global protein expression by median centering across 151 antibodies tested. Individual protein expressions in DSRCT specimens, ES specimens, and mesenteric normal tissue specimens were compared using two-sided unpaired $t$-tests with the GeneSpring GX software program version 12.6.1-GX (Agilent Technologies). For multiple comparison testing, proteins whose expression levels between specimen types were two-fold and significantly different ($p \leq 0.05$) were subjected to unsupervised hierarchical clustering. Differentially expressed proteins identified by RPPA were confirmed by WB analysis. The RPPA data for the proteomic comparison of DSRCT patient tumors, normal mesenteric tissues, and ES patient tumors are available from the Gene Expression Omnibus repository under GSE108687 series.

The preparation of extracted protein from cells, DSRCT patient tumors, or xenograft animal tumors for western blotting validations was prepared as described previously[62]. Lysis buffer (1% Triton X-100, 50 mM HEPES, pH 7.4, 150 mM NaCl, 1.5 mM MgCl₂, 1 mM EGTA, 100 mM NaF, 10 mM Na pyrophosphate, 1 mM Na₃VO₄, 10% glycerol) containing freshly added protease and phosphatase inhibitors (Roche Applied Sciences) was applied to lyse cellular washed pellets via cold incubation. Next, protein extraction from xenograft tumors was performed by homogenizing approximately 10 mg of frozen tissue in 500 μL of the lysis buffer using an electric tissue homogenizer (PRO Scientific). The homogenized tumors were incubated at +4 °C for 2 h to complete their dissociation and lysis. Altogether, the total lysed proteins from tumors were collected after centrifugation, quantified using BCA protein assay kit (Thermo Fisher Scientific), and stored at −80 °C until further analyses. WBs were performed as previously described by our group[61,62]. Protein detection was performed using a list of antibodies provided in the above supplementary table of antibodies used for WBs. The immune-reactive proteins were captured using horseradish peroxidase-conjugated secondary anti-rabbit IgG or anti-mouse IgG antibodies (Cell Signaling Technology), amplified using the SuperSignal West Dura chemiluminescent substrate (Thermo Fisher Scientific), detected using the Chemi-Doc system (Bio-Rad), and quantified for their densitometry using the ImageJ Gel Analysis tool (NIH, Bethesda, MD).

**TMA preparation and immunohistochemistry analyses**. Four-micrometer unstained slides were prepared from this TMA. Immunohistochemical studies were performed using an autostainer (Bond-Max; Leica Microsystems, Buffalo Grove, IL, USA) with anti-AR (1:30; clone AR441, Dako#M3562) antibody. This primary antibody was detected using a Bond polymer refine detection kit according to the manufacturer protocol (Leica, #DS9800). Then, the slides were dehydrated in grade alcohols, cleared in xylene, and cover slipped. The TMA staining results were scored on intensity (low, moderate, and high) as well as the percentage of tumoral labeling (positive (>50%), low positive (10–50%), focal (1–10%), and negative (0–1%)).

**WST1 cell proliferation assays**. The cells were cultivated for 24 h to let them adhere, before adding serially diluted concentrations of dihydrotestosterone (DHT, 0.0064-2000nM; Selleckchem), or AR-based drug treatments (Enzalutamide, AR-ASO; 0.032–5 mM) for up to 48 or 72 h, respectively. WST1 was added, and cells were incubated for an additional 2 h. Cell proliferation was measured at 450 nm in a microplate reader (DTX880, Beckman Coulter). In both assays, the agonistic effect of DHT or cytotoxic effect of the drugs was expressed as a percentage of cell proliferation. IC50 values were calculated by the sigmoidal dose-response curve-fit using Prism GraphPad 8.0.

**Immunohistochemistry and digital image analyses**. Primary tumors from the DSRCT xenograft and PDX1 group of mice, untreated and treated with control-ASO or AR-ASO, were fixed in 10% formalin, embedded in paraffin, and then sliced in 5-μm sections. The EZ-retriever microwave-based pretreatment and antigen retrieval system (Biogenex, CA) was used for dewaxing, rehydration, and antigen retrieval of these FFPE lung tissue sections. AR protein expression by IHC was evaluated on Leica Bond MAX Autostainer by using primary antibody, rabbit antibody anti-human AR (Cell Signaling Technologies, #5153). The primary antibody was detected according to the manufacturer's protocol (Leica, #DS9800). Then, the slides were dehydrated in grade alcohols, cleared in xylene, cover slipped, and imaged with Keyence Microscope (Keyence, Tokyo, Japan) at ×20 resolution. The digital images were processed and quantified using the Visiopharm software version (2020.04) (Hoersholm, Denmark). An APP (Analysis Protocol Package) was designed to quantify cell-based DAB staining in DSRCT preclinical samples using traditional thresholding methods. The algorithm was built around a three-step approach: (1) pre-processing: HDAB-DAB and HDAB-hematoxylin features were used to detect AR-positive cells (green masking) and negative cells (blue masking), respectively, by setting pixel values. A median filter of size 5 × 5 was used for the proper segmentation of cells. (2) Post processing: additional steps were designed to enhance the performance of the APP. Change by shape excludes the artifacts; merged cells were separated by separate labels step, and certain clear areas in the cells were filled with fill holes step to fully mask the cells. (3) Output variable: mean intensity of each cell and from each sample was extracted and exported into a spreadsheet. Finally, the data were plotted on a Scatter Plot using GraphPad Prism software, version 8 showing the AR Mean Intensity of each cell from all the analyzed samples, and the AR Average mean intensity from each preclinical group of treated mice.

**ChIP-seq data processing**. ChIP-seq data were quality controlled and processed by pyflow-ChIP-seq[71], a snakemake[72] based ChIP-seq pipeline. Briefly, raw reads were mapped by bowtie1[73] to hg19. Duplicated reads were removed, and only uniquely mapped reads were retained. RPKM normalized bigwigs were generated by deep tools[74], and tracks were visualized with IGV[75]. Peaks were called using macs1.4[76] with a p value of 1e−9 for H3K27ac and 1e−7 for AR. Heatmaps were generated using R package *EnrichedHeatmap*. ChIP-seq peaks were annotated with the nearest genes using ChIPseeker[77]. SEs were identified using *ROSE*[78] based on H3K27ac ChIP-seq data.

**Differential peaks analysis**. To identify variable AR or enhancer domains enriched in specific DSRCT samples, enhancer peaks that overlap with 2.5 kb upstream and 2.5 kb downstream of any known TSSs were removed. The unique and shared peaks within multiple groups were identified by Intervene[79]. The peaks were annotated with ChIPseeker R package[77], using addFlankGeneInfo function for enhancers.

**Identification of AR and enhancer associated pathways**. Differential AR binding sites and enhancers associated genes in each sample were imported into the ClusterProfiler[80] for pathway analysis, restricted to GO, KEGG, Hallmark, and WiKi gene sets. The Enrichplot package[81] was used to generate dot plot and bar plot for gene sets enriched with a false discovery rate cut-off of <0.05.

**Enrichment of motifs in AR-specific peaks**. To identify the motifs over-represented within AR-specific peak sets, we used the HOMER motif database and the coordinates of AR-specific peak sets[82].

**Statistics and reproducibility**. Bar and scatter dot plots are presented as mean values ± SD. Box plots indicate the median value and interquartile range; whiskers show minimum and maximum data points. All *t*-test p values are two-sided. To enhance reproducibility, no less than five animals were assigned per group. Three

or more regions of interest were randomly assessed for each microscopy image, and representative images are shown.

**Reporting summary**. Further information on research design is available in the Nature Research Reporting Summary linked to this article.

## Data availability

All data that support the findings of this study are provided within the article, Supplementary Information, and source data. All ChIP-seq data are available at GEO accession number GSE151380, released publicly on September 1, 2022. RNA-seq data are made publicly available in the European Genome-Phenome Archive: DSRCT at EGAS00001004575; liposarcoma at EGAS00001002807; osteosarcoma at EGAS00001003247; chondrosarcoma at EGAS00001004585; and prostate cancer through Subudhi et al.[83]. RPPA DSRCT GEO (GSE108687) RNA-Seq DSRCT EGAS00001004575 [https://ega-archive.org/datasets/EGAD00001006394]. RNA-Seq LS data EGAS00001002807. RNA-Seq OS data EGAS00001003247. RNA-Seq CS data EGAS00001004585. RNA-Seq for PC: EGAS00001004050. RPPA data GEO (GSE178406). AR-ChIP-seq GEO (accession #: GSE151380). H3K27Ac ChIP-seq GEO (accession #: GSE151380). Additional RPPA GEO (accession #: GSE178406).

## Code availability

The code used to generate the ChIP-seq data are available at https://github.com/crazyhottommy/pyflow-ChIPseq.

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

## Acknowledgements

Investigators at The University of Texas MD Anderson Cancer Center are supported by the National Institutes of Health through Cancer Center Support Grant CA016672. The Characterized Cell Line Core Facility is funded by the NIH [CA016672]. K.R. and M.M. were supported by grants from the NCI (CA222214, CA245395, CA226269), Cancer Prevention Research Institute of Texas (CPRIT; RP220410 and RP200390), and MDACC intramural funds. P.A.F. is supported by the Cancer Prevention Research Institute (R120501) and the Welch Foundation's Robert A. Welch Distinguished University Chair Award (G-0040). We thank the Cory Monzingo Foundation and Blake Abercrombie Foundation for their generous philanthropy that supported this work.

## Author contributions

S.-E.L.-C., M.M., B.A.M., S.K., A.R.V., P.C., R.W.P., V.R., S.M., D.D.T., A.R.V., M.T., J.A.Lu., W.-L.W., and C.R.-.A. designed experiments, performed experiments, and analyzed data. D.R.I. helped identify and collect clinical samples for analysis. S.-E.L.-C., J.A.Lu., A.H.-J., and K.R. analyzed data and wrote the manuscript. P.A.F., M.T., A.R.M., N.C.D., R.R., J.A.Li., and A.H.-J. provided key scientific input and expertise. S.-E.L.-C., N.C.D., B.C., D.M., A.H.-J, R.R., J.A.Li., and J.A.Lu. helped identify the enrolled DSRCT patients in the study. B.U. assisted with confocal microscopy. W.-L.W. and A.J.L. provided pathology interpretations. C.R.-A. assisted with preclinical in vivo experiments. M.T. performed the mass spectroscopy analysis. C.-C.W., H.C.B., and P.A.F. performed the transcriptomic analysis. N.E.A., M.M., and K.R. performed the ChIP-seq experiments and aided in their interpretation. All authors participated in data interpretation and manuscript editing.

## Competing interests

A.R.M. is an employee and shareholder of Ionis Pharmaceuticals. The remaining authors declare no competing interests.
