## [Peer Review File · Nature Communications]

The androgen receptor is a therapeutic target in desmoplastic small round cell sarcomaEditorial Note: Parts of this Peer Review File have been redacted as indicated to remove third-party material where no permission to publish could be obtained.

REVIEWER COMMENTS

Reviewer #1 (Remarks to the Author):

This study highlights AR as a potential therapeutic target in DSRCT. Given that AR-targeted FDA-approved agents are already available, the findings are of immediate clinical interest.

1. General comment: the emphasis in the pre-clinical in vivo studies on the AR ASO (AZD5312) with so far only Phase 1 data known, over the widely available FDA-approved enzalutamide, is unfortunate. It also begs the higher level question: if one were to treat DSRCT with an ASO, why not design one specific to the EWSR1-WT1 fusion product?
2. General comment: while the male preponderance of DSRCT is striking, there are nonetheless about 10-15% of cases in women. Obviously, most do not express high levels of AR, as shown in Fig 2A. If androgens promote DSRCT growth, as shown in Fig 4A, one might postulate that DSRCT is less aggressive in women but, to my knowledge, this is not the case. The authors should discuss this apparent paradox. The same question could be raised about the approximately 10% of DSRCT that arise pre-pubertally.
3. General comment: the authors should compare their RPPA analysis which uncovered AR expression to the recently published whole transcriptome analysis in which an AR transcriptional signature does not appear among their "Significant differentially enriched gene sets in DSRCT" (PMID: 32703985). The potential technical or biological reasons for the discrepancy should be discussed.
4. The Intro is way too long, more akin to the background section of a PhD thesis. Paragraphs 3, 4, and 5 are mostly non-essential information and could easily be removed.
5. The intro tries to provide some historical background but misstates it. " ...it wasn't until 1989 that Gerald and Rosai recognized a pathognomonic t(11;22)(p13;q12) chromosomal translocation that pairs the Ewing sarcoma (ES) gene (EWSR1) with the Wilm's tumor suppressor gene (WT1)" is incorrect. Ditto for opening sentence of Discussion. The correct chronology is:
 - a. Clinicopathologic description: Gerald & Rosai 1989, 1991
 - b. Cytogenetic reports of t(11;22)(p13;q12): 4 groups, incl. Gerald, 1992-93
 - c. Identification of EWS-WT1 fusion: Ladanyi & Gerald, 1994 (PMID: 8187063)
6. Wilms tumor, not "Wilm's".
7. "in a recent review by Mora and Rosai" should be corrected to "in a recent review by Mora et al" (there were 9 other co-authors).
8. Fig 2A: if outcome data are available for the patients in the TMA, an analysis of whether AR expression level correlates with survival would be of great interest.
9. Fig 3: I find the PSA analysis very weak. Why a comparison to only 3 ES samples, of which one is pre-pubertal? Why are there 3 ES samples in panel A but 6 in panel B? Panel A is insufficient to convince readers of a meaningful difference between DSRCT and ES and the p value <0.0001 in Panel B is hard to believe with so few samples in one of the two groups. I suggest that the authors substantially expand this analysis or save it for a future paper.

Reviewer #2 (Remarks to the Author):

This is a relatively comprehensive preclinical study of androgen receptor inhibition (mainly by AR ASO) in DSRCT. The putative role of AR in this disease was reported previously (e.g., Fine et al, 2006). The stated main motivation for revisiting the role of AR was the availability of approved and experimental anti-AR drugs (including AR ASO). Overall, the study was enabled by a large collection of clinical specimens and the amount of the data presented in the study is highly appreciated. However, the notion that AR is a valid target in DSRCT does not appear to be fully supported by the data. AR is generally expressed in many tissue types but as a disease driver only validated in prostate cancer. Although it has been implicated in other cancer types (e.g., triple negative BC, salivary duct carcinoma, DSRCT, some brain tumors), it is a big hurdle requiring a robust study design in order to validate AR as a disease driver for the purpose of justifying a trial in DSRCT. The following critiques are contributed to the authors for consideration. They are organized according to the sequence of the Figures presented.

Figure 1 and Supplemental Figure 1 (protein lysate analysis): It should be clarified how the

samples were selected and whether selection biases exist as the specimens appear to be fresh frozen samples. It is interesting that DSRCT samples have higher AKT activity but its relationship to AR signaling was not evaluated. In prostate cancer, the two pathways are reciprocally regulated, i.e., suppression of one would activate the other. This is probably the most important question to examine in light of the interest in treating the tumors by AR suppression. But it is not discussed in the paper, though the data in Figure 6A validates AKT activation in ASO treated tumors.

Figure 2 and Supplemental Figure 2 (TMA and gene expression analysis): For TMA, it should be clarified how AR positivity was defined and how values from TMA cores from the same patients were tabulated in the analysis. It was stated 65% showed high AR expression. However given DSRCT has multiple histologies within the same patient it is highly unlikely that AR would be uniformly positive across lesions as seen in prostate cancer--this is clinically relevant (in addition to the reciprocal regulation mentioned above) and may explain minimal clinical benefit reported in Fine et al. For expression analysis, it should be clarified how samples were selected. For example, it was shown the AR levels in DSRCT were substantially lower when compared to prostate cancer but more information on the prostate cancer specimens should be provided because AR expression varies quietly substantially between primary tumor and tumors progressing on AR suppression.

Figure 3 (PSA ELISA): Although PSA tests in serum samples (all male pts) were performed using a validated ELISA kit, it is not certain whether the values are comparable with clinical PSA tests. Negative controls as well as controls from patient sample with known PSA values may help to calibrate the assay. As it is presented it is not convincing that this group of 20 young men (including the 3 ES pts) would all have PSA greater than 10ng/ml.

Figure 4 (DHT stimulated growth and growth suppression by AR suppression): The experiments in general could have been better controlled with addition of responsive versus nonresponsive cell lines included in all experiment conditions, and with experimental conditions including with/without DHT conditions. It is not clear whether AR nuclear localization definitively took place upon DHT, and there was no indicator of AR activation (e.g., PSA) used in any of the experiments. AR protein should appear as a single band in LNCAP cells (unless there is degradation).

Figure 5&6 (preclinical studies): Ideally enzalutamide treatment should be used consistently in both cell line and PDX to enable robust interpretation of the data.

Figure 7&8 (genome-wide AR binding): data presented in the two figures adds very little to the overall study and could be curtailed.

Reviewer #3 (Remarks to the Author):

In this study the authors study an ultra-rare sarcoma subtype and seeks to demonstrate that AR is a targetable dependency in DSRCT. The authors has examined a relatively large number of cases by RPPA and identified AR as a candidate target in this disease. However, in its current form, the manuscript lacks many details and is preliminary which needs to be addressed before further consideration.

Major points:

1. In general the quality of the majority of figures is very poor. For instance, the detail in most heatmaps is difficult to see as font is too small.

2. All patient demographic data from frozen specimens and TMA are missing. Are the tissue primary, metastatic, biopsies or resections, treatment naïve or prior treatment? This is particularly important when comparing RPPA and AR data between different DSRCT specimens. Please provide this information in a demographic table.

3. There is a lack of consistency in the blots/RPPA data, e.g. 13T and 13N are found in blots but not RPPA data in Figure 1 and S1?

4. What is the AR expression different between DSRCT and normal? This data is not provided in Figure S1
5. All replicate information need to be provided, e.g. RPPA and western blot quant, no replicate information in provided in the figure legends. If western blots are n=1, replicates need to be performed prior to quantitation
6. Figure S2D and S2E are too small to see anything. Also not obvious from heat map that these are really differentially expressed in DSRCT vs rest, how was DEG determined? E.g. by SAM? If they do unsupervised clustering, do the DSRCT cases cluster together based on these DEGs. Can they provide the GSEA/pathway enrichment data to demonstrate that AR pathway is enriched?
7. Figure 4 Figure replicates information missing for A, B, E, F, G. Also 4E and F, stats are missing, so how can one say the data is significant. 4E&F and S4 are dose response assays based on the cell viability dye, it is inappropriate to say that it "significantly slowed proliferation" as it is a composite readout of that is a combination of cell growth, senescence and death. It would be % viability. If the authors want to report proliferation, count the cells.
8. Figure 5, error bars for tumour curves and stats for KM curves are missing.
9. Important information about the DSRCT PDX model is missing. How does it compare to the human patient specimen in terms of histology and molecular readout? How many passages from original implantation. Patient information is again missing from this, is this primary specimen, relapse, treatment naïve etc?
10. Fig 6A is difficult to see, e.g early pharmacodynamics changes, e.g pS6 etc are they significantly different?
11. There is an inherent challenge with working with rare entities that good preclinical models are lacking. However it is also important that any findings reported are robust. It would be critical to see that the data in figure 7 and 8 are reproduced using the DSRCT PDX model. If the authors are not able to generate short-term primary cultures from the PDX tumours to do these experiments, then I would suggest doing them in PDX tumour explants. It is insufficient to just rely on data from one immortalised cell line to assert conclusions regarding AR biology in DSRCT.

Other points

1. This statement is speculation not fact "Though the DSRCT FP shares an N-terminal EWSR1 gene with ES, the tendency to arise within the abdomen must be driven mainly by the aberrant C-terminus WT-1 protein and its downstream epigenetic effects."
2. Figure S1 only shows a subset of proteins in RPPA, which does not support the statement "unsupervised HC correctly separate normal from DSRCT" as specific proteins were chosen, how were these proteins chosen, please show full datasets.
3. By definition RPPA is not unbiased, it is a targeted approach because antibody targets are pre-selected, e.g. unlike mass spectrometry. Please amend the text.
4. Figure 1B, D, please replace biomarkers on x axis with proteins. These are not biomarkers unless independently validated
5. Figure 2 – consistency of figure and text. Text use high while figure uses positive. What is the cut-off used for IHC and WB
6. For PC samples use in Fig 2E-F, what type were they, treatment naïve or have undergone treatment.
7. Figure 3B stats, it is not appropriate to treat technical replicates as different points for the same

patient in a dot blot, should use the mean of the two technical replicates.

8. This statement "suggest a novel AR-dependent integrin/NCOA-dependent pathway exists in DSRCT that might trigger DSRCT cell migration and death" – Not supported by integrin data, which they claim is not associated with AR expression? How did they extrapolate this to cell migration and death? Very speculative statement not supported by data

9. Is Fig 4G all from the same blot? At the moment it looks like it is spliced from a different blot. Again replicates for blot quant is missing

10. RPPA data is not provided/deposited

REVIEWER COMMENTS

Reviewer #1:

This study highlights AR as a potential therapeutic target in DSRCT. Given that AR-targeted FDA-approved agents are already available, the findings are of immediate clinical interest.

Author response: We thank reviewer #1 for especially constructive comments that have helped us significantly improve the quality of data and readability of our manuscript. We agree, the results have immediately clinical impact and we are, in fact, designing a trial to test enzalutamide or AZD5312 prospectively.

1. General comment: The emphasis in the pre-clinical in vivo studies on the AR ASO (AZD5312) with so far only Phase 1 data known, over the widely available FDA-approved enzalutamide, is unfortunate. It also begs the higher level question: if one were to treat DSRCT with an ASO, why not design one specific to the EWSR1-WT1 fusion product?

Author response: The reviewer's comment is well received. Our initial focus regarding the AR-ASO (now in phase 2) was both pragmatic and scientifically-based. To explain...our institution treats ~20 newly diagnosed DSRCT patients each year and, to date, we haven't had success getting insurance companies to provide enzalutamide off-label for our patients. We hope the data presented in our manuscript, now that it includes additional data using enzalutamide, will change that once published. Furthermore, given the rarity of DSRCT and lack of perceived return on investment, Pfizer was unwilling to provide free enzalutamide for prospective clinical evaluation in this sarcoma subtype nor willing to fund a clinical trial to study a second potential cancer indication.

In contrast, Ionis Pharmaceuticals was already working with MD Anderson to develop a YAP1-selective ASO now in phase 1 trials (NCT04659096) and exploring an EWSR1-targeted ASO preclinically that might exhibit antineoplastic activity in Ewing sarcoma, DSRCT, and other fusion-positive sarcoma subtypes that also harbor an oncogenic N-terminus EWSR1 gene. Given this relationship, they have tentatively agreed to fund a DSRCT-specific phase 2 trial to investigate their AR-ASO, with or without enzalutamide.

As the reviewer notes, an ASO targeting WT1 might also work, but Ionis wouldn't commit to developing an WT1-selective ASO that would theoretically work only in one pediatric cancer type. As one looks at our data, the AR-ASO appears to be slightly more effective than enzalutamide and works differently than enzalutamide and other FDA-approved cancer drugs by eradicating AR levels rather than merely regulating AR function. Given its unique MOA, our manuscript provides the preclinical foundation to develop a new drug candidate in DSRCT while providing novel insights regarding enzalutamide that might facilitate greater off-label access.

A final point to make, is that other companies are developing AR-targeted therapies that work by reducing AR protein levels. Though we didn't have access to the class of AR PROTACs (e.g., ARV-110 and ARV-766) now in phase 2 trials by ARVINAS, it hasn't escaped us that their investigational agents may act similarly to an AR-ASO.

2. General comment: While the male preponderance of DSRCT is striking, there are nonetheless about 10-15% of cases in women. Obviously, most do not express high levels of AR, as shown in Fig 2A. If androgens promote DSRCT growth, as shown in Fig 3A, one might postulate that DSRCT is less aggressive in women but, to my knowledge, this is not the case. The authors should discuss this apparent paradox. The same question could be raised about the approximately 10% of DSRCT that arise pre-pubertally.

Author response: We thank the reviewer for this astute comment. Given the rarity of female or prepubertal DSRCT patients, we can't, with any confidence, determine what instigates or sustains DSRCT in this atypical patient population. Nevertheless, we can draw lessons from other hormonally-driven malignancies and speculate based upon our data why females might develop this disease.

First regarding gender, there's no denying that estrogen drives breast cancer in the vast majority of female patients, and yet male patients can also develop this. As with male breast cancer patients, we can only surmise that other factors contribute to tumorigenesis in female or pre-pubertal DSRCT patients. Though we don't have enough female patients in our study to answer conclusively, and therefore wouldn't want to speculate formally in our manuscript, one hypothesis is that female DSRCT tumor are estrogen-driven. It's well reported that ER, AR, and other steroid receptor hormone family members have shared DNA binding motifs, with the major hormonal effects dictated both by the intratumoral hormone levels and tissue-specific effects of hormones within different tissues or organs.

3. General comment: The authors should compare their RPPA analysis which uncovered AR expression to the recently published whole transcriptome analysis in which an AR transcriptional signature does not appear among their “significant differentially enriched gene sets in DSRCT” (PMID: 32703985). The potential technical or biological reasons for the discrepancy should be discussed.

Author response: We thank the reviewer for referring us to this important publication. Though an AR transcriptomic signature wasn't among the top gene signatures identified in DSRCT, we believe protein levels generally provide a more direct impact on cell phenotype since they're directly involved in modulating cell behavior. Not only can transcript levels poorly correlate with protein expression, often times a protein's effect depends on its phosphorylation status, its intracellular location, and its binding to various co-factors. This is certainly true for AR, whose complex genome-wide epigenetic effects depends upon ligand-binding to promote nuclear shuttling and numerous co-factors that alter DNA binding.

For these reasons, we used RPPA to analyze patient and preclinical animal samples, using 210 high-quality antibodies that target total proteins and post-translationally modified proteins. These antibodies cover a wide range of oncogenic functions that affect cell proliferation, DNA damage, cell polarity, vesicle function, EMT, invasiveness, hormone signaling, apoptosis, metabolism, immunological and stromal function, as well as transmembrane myriad oncogenic signaling pathways. In addition to the interesting work by Hingorani et al., we are eager to compare our results other -omic studies for DSRCT that they emerge in the future.

4. General comment: The Intro is way too long, more akin to the background section of a PhD thesis. Paragraphs 3, 4, and 5 are mostly non-essential information and could easily be removed.

Author response: We thank the reviewer for this recommendation, agree, and have significantly trimmed our Introduction section by removing paragraphs 3-5. As revised, we get to the point quickly and focus squarely at the topic at hand.

5. Specific Comment: The intro tries to provide some historical background but misstates it. “... it wasn't until 1989 that Gerald and Rosai recognized a pathognomonic t(11;22)(p13;q12) chromosomal translocation that pairs the Ewing sarcoma (ES) gene (EWSR1) with the Wilm's tumor suppressor gene (WT1)” is incorrect. Ditto for opening sentence of Discussion. The correct chronology is:

- a. Clinicopathologic description: Gerald & Rosai 1989, 1991
- b. Cytogenetic reports of t(11;22)(p13;q12): 4 groups, incl. Gerald, 1992-93
- c. Identification of EWS-WT1 fusion: Ladanyi & Gerald, 1994 (PMID: 8187063)

Author response: Very good point regarding the precise historical context of the clinicopathologic, cytogenetic, and genetic discoveries. We have edited our manuscript accordingly and updated the citations with those provided by the reviewer.

6. Specific Comment: Wilms tumor, not “Wilm's”.

Author response: Corrected. Thanks for catching this grammatical error.

7. Specific Comment: “in a recent review by Mora and Rosai” should be corrected to “in a recent review by Mora et al” (there were 9 other co-authors).

Author response: This clause was removed entirely while trimming the Introduction.

8. Regarding Fig 2A: If outcome data are available for the patients in the TMA, an analysis of whether AR expression level correlates with survival would be of great interest.

Author response: Unfortunately, we had insufficient outcomes data to include outcomes data within the current manuscript. We agree it's an incredibly interesting question, and one that will hopefully get answered prospectively within the confines of a DSRCT-specific AR-direct clinical trial. We note that her2/neu was originally an adverse prognostic biomarker in breast cancer until the advent of Herceptin and suspect a similar story might play out in DSRCT to the extent we can target AR with an AR-ASO, enzalutamide, or another prostate cancer oriented drug.

9. Regarding Fig 3: I find the PSA analysis very weak. Why a comparison to only 3 ES samples, of which one is pre-pubertal? Why are there 3 ES samples in panel A but 6 in panel B? Panel A is insufficient to convince readers of a meaningful difference between DSRCT and ES and the p value <0.0001 in Panel B is hard to believe with so few samples in one of the two groups. I suggest that the authors substantially expand this analysis or save it for a future paper.

Author response: We agree with the reviewer comment and have decided to shift the PSA analysis for a future manuscript.

* * *

Reviewer #2

This is a relatively comprehensive preclinical study of androgen receptor inhibition (mainly by AR ASO) in DSRCT. The putative role of AR in this disease was reported previously (e.g., Fine et al, 2006). The stated main motivation for revisiting the role of AR was the availability of approved and experimental anti-AR drugs (including AR ASO). Overall, the study was enabled by a large collection of clinical specimens and the amount of the data presented in the study is highly appreciated. However, the notion that AR is a valid target in DSRCT does not appear to be fully supported by the data. AR is generally expressed in many tissue types but as a disease driver only validated in prostate cancer. Although it has been implicated in other cancer types (e.g., triple negative BC, salivary duct carcinoma, DSRCT, some brain tumors), it is a big hurdle requiring a robust study design in order to validate AR as a disease driver for the purpose of justifying a trial in DSRCT. The following critiques are contributed to the authors for consideration. They are organized according to the sequence of the Figures presented.

Author response: As the reviewer correctly points out, our comprehensive research takes just one of many steps that will be required to validate AR as a clinically relevant target in DSRCT. We are indebted to this reviewer for his or her in-depth comments that track with each figure and have made all requested changes. The updated version is now significantly improved based upon these constructive comments.

Figure 1 and Supplemental Figure 1 (protein lysate analysis): It should be clarified how the samples were selected and whether selection biases exist as the specimens appear to be fresh frozen samples. It is interesting that DSRCT samples have higher AKT activity but its relationship to AR signaling was not evaluated. In prostate cancer, the two pathways are reciprocally regulated, i.e., suppression of one would activate the other. This is probably the most important question to examine in light of the interest in treating the tumors by AR suppression. But it is not discussed in the paper, though the data in Figure 6A validates AKT activation in ASO treated tumors.

Author response: We amended the manuscript within the “Patients and Methods” section to clarify how we selected samples for RPPA and TMA analyses, updated the histograms in the Figure 1, and the corrected supplemental Figure 1.

We thank the reviewer for highlighting the reciprocal relationship between AR and Akt signaling, which we were unfamiliar with. Based upon this astute observation in our data, we have added several sentences to the Discussion section, raising the interesting possibility that the IGF/PI3K/mTOR pathway targeting might complement AR-directed therapy. We and others have noted some activity of temsirolimus in DSRCT, so it wouldn't be a stretch in future research to explore cross-talk between the AR and IGF/PI3K/mTOR pathway.

Figure 2 and Supplemental Figure 2 (TMA and gene expression analysis): For TMA, it should be clarified how AR positivity was defined and how values from TMA cores from the same patients were tabulated in the analysis. It was stated 65% showed high AR expression. However, given DSRCT has multiple histologies within the same patient it is highly unlikely that AR would be uniformly positive across lesions as seen in prostate cancer--this is clinically relevant (in addition to the reciprocal regulation mentioned above) and may explain minimal clinical benefit reported in Fine et al. For expression analysis, it should be clarified how samples were selected. For example, it was shown the AR levels in DSRCT were substantially lower when compared to prostate cancer but more information on the prostate cancer specimens should be provided because AR expression varies quietly substantially between primary tumor and tumors progressing on AR suppression.

Author response: We thank the reviewer for this constructive comment. We updated the “Expression of AR in DSRCT primary tumors” results section and Figures 2B-2D to more clearly specify how AR positivity was defined. The AR positivity was defined based on the percentage of cells that are positive for AR expression. The patient tumors are either high (>50% cells), low (10-50% cells), focal (1-10% cells), or negative (0-1% cells). We've also

noted that the TMA was created from all available DSRCT tumors that exist within MD Anderson without selection for any demographic or biomarker trait.

Prior Figure 3 (PSA ELISA) – now deleted: Although PSA tests in serum samples (all male pts) were performed using a validated ELISA kit, it is not certain whether the values are comparable with clinical PSA tests. Negative controls as well as controls from patient sample with known PSA values may help to calibrate the assay. As it is presented it is not convincing that this group of 20 young men (including the 3 ES pts) would all have PSA greater than 10ng/ml.

Author response: Reviewer #2 raises a valid question also echoed by Reviewer #1, who suggested that the PSA data should be moved to a different manuscript. We agree, have removed Fig. 3, and have just recently begun to test DSRCT patients prospectively for PSA within our institution's CLIA-certified clinical laboratory.

Figure 3 (DHT stimulated growth and growth suppression by AR suppression): The experiments in general could have been better controlled with addition of responsive versus nonresponsive cell lines included in all experiment conditions, and with experimental conditions including with/without DHT conditions. It is not clear whether AR nuclear localization definitively took place upon DHT, and there was no indicator of AR activation (e.g., PSA) used in any of the experiments. AR protein should appear as a single band in LNCaP cells (unless there is degradation).

Author response: We thank the reviewer for the constructive recommendation. Accordingly, we repeated the experiment using responsive (JN-DSRCT and LNCaP) versus nonresponsive (TC71 and PC3) cell lines. Additionally, we performed confocal microscopy for AR expression within all selected cell lines, in the absence or presence of DHT, to obtain higher quality images. Expectedly, DHT stimulation facilitated AR transmigration from the cytoplasm into the cell nucleus within LNCaP and JN-DSRCT cells, which highly express the AR biomarker, but not in PC3 and TC-71 cells that lack AR expression.

Regarding the double bands observed for AR expression in LNCaP, we can't exclude the possibility of limited AR degradation but routinely detect two bands using the highly cited polyclonal antibody (#5153) purchased from Cell Signaling Technology (CST). In our experience, we routinely see a strong upper band and weaker lower band assessing LNCaP by Western blot. Our results are consistent with the datasheet provided by CST, which also shows the double band. See attached figure from their website (<https://www.cellsignal.com/products/primary-antibodies/androgen-receptor-d6f11-xp-rabbit-mab/5153?site-search-type=Products&N=4294956287&Ntt=ar&fromPage=plp>).

[REDACTED]

Figure 4 & 5 (preclinical studies): Ideally enzalutamide treatment should be used consistently in both cell line and PDX to enable robust interpretation of the data.

Author response: We thank the reviewer for this suggestion and have amended Fig. 4 to include an additional DSRCT PDX treated with enzalutamide. The results obtained with this new PDX are in close agreement with the promising JN-DSRCT xenograft data.

Figure 6 & 7 (genome-wide AR binding): data presented in the two figures adds very little to the overall study and could be curtailed.

Author response: Having added an additional DSRCT model to our analysis, we believe the additional epigenetic information gained from the ChIP-seq analysis provides valuable hypothesis-generating information that the readers will find informative. At the suggestion of Reviewer #3, we have kept figures 6 & 7 and repeated the same analyses using the preclinical samples.

Reviewer #3:

In this study the authors study an ultra-rare sarcoma subtype and seeks to demonstrate that AR is a targetable dependency in DSRCT. The authors has examined a relatively large number of cases by RPPA and identified AR as a candidate target in this disease. However, in its current form, the manuscript lacks many details and is preliminary which needs to be addressed before further consideration.

Author response: We thank the reviewer for the thorough review of our manuscript and invaluable feedback that has allowed us to significantly strengthen its clarity, conciseness, and overall impact.

Major points:

1. In general, the quality of the majority of figures is very poor. For instance, the detail in most heatmaps is difficult to see as font is too small.

Author response: We thank the reviewer for raising this point and have redone all the figures, both to increase their resolution and increase the font size to improve legibility.

2. All patient demographic data from frozen specimens and TMA are missing. Are the tissue primary, metastatic, biopsies or resections, treatment naïve or prior treatment? This is particularly important when comparing RPPA and AR data between different DSRCT specimens. Please provide this information in a demographic table.

Author response: We thank the reviewer for this very constructive recommendation. Figure 2A now includes key demographic data, including gender, timing of the biopsy used for the TMA, and information detailing whether the specimen was collected before or after chemotherapy. Supplemental Table 4 summarizes the demographic information of DSRCT and ES patients that are used in the RPPA analysis.

3. There is a lack of consistency in the blots/RPPA data (e.g., 13T and 13N are found in blots but not RPPA data in Figure 1 and S1)?

Author response: We thank the reviewer for identifying this error. We inadvertently mislabeled samples 13T and 13N on Figure S1 and have updated this figure with the correct sample labels (15T and 15N). We did not include sample 13T on S1 because the corresponding adjacent normal tissue sample (13N) wasn't collected from that particular patient.

4. What is the AR expression different between DSRCT and normal? This data is not provided in Figure S1

Author response: We thank the reviewer for this thoughtful recommendation and have updated Figure S1 (panel B) to include the RPPA profile for AR expression. AR remains statistically significantly higher in the DSRCT tumor tissue compared to the and surrounding normal tissue (p -value = 0.02).

5. All replicate information need to be provided (e.g., RPPA and western blot quants); no replicate information is provided in the figure legends. If western blots are $n=1$, replicates need to be performed prior to quantitation

Author response: The reviewer raises a valid point and we have updated the manuscript accordingly. Wherever appropriate, we have updated our figures to indicate where groups or conditions are statistically significant. We have also updated the figure legends with information describing the number of experimental replicates. Please see updated Figures 1, 3, S1, and S6.

6. Figure S2D and S2E are too small to see anything. Also, it's not obvious from heat map that these are really differentially expressed in DSRCT vs rest, how was DEG determined? E.g., by SAM? If they do unsupervised clustering, do the DSRCT cases cluster together based on these DEGs. Can they provide the GSEA/pathway enrichment data to demonstrate that AR pathway is enriched?

Author response: We thank the reviewer for this important comment and have updated figures S2D and S2E to improve readability. The unsupervised clustering was performed using *all genes* present in the GO AR signaling and KEGG prostate cancer pathways, not a subset of DEG. The DSRCT samples tended to cluster together but were not distinctly separate from other sarcoma subtypes. Therefore, the degree of activation of these pathways were not higher in DSRCT as compared to other sarcoma subtypes. GSEA was used to determine the level of deregulation between DSRCT and the other sarcoma subtypes using gene expression fold changes between the

two groups. Their p-values were not significant for these pathways ($p < 0.08$ and $p < 0.234$, respectively). Our interpretation is that AR activation induces a slightly different transcriptomic signature in DSRCT compared to prostate cancer. This result is expected for two reasons: (1) AR likely induces tissue/organ-specific effects (as observed for breast and prostate tissues), and (2) the epigenetic effects of EWS-WT1 fusion protein almost certainly alters the enhancer/promoter DNA binding pattern of AR.

7. Figure 3: Replicates information missing for A, B, E, F, G. Also, 3E and F, are missing stats, so how can one say the data is significant. 3E&F and S4 are dose response assays based on the cell viability dye, it is inappropriate to say that it “significantly slowed proliferation” as it is a composite readout of that is a combination of cell growth, senescence, and death. It would be % viability. If the authors want to report proliferation, count the cells.

Author response: Figure 3 has been updated to add three replicates for each cell line and western blot sample. Additionally, we have annotated the figure with the statistical information needed so readers can correctly compare treatment groups. Figures 3A, 3G (old 3E), and 3H (old 3F) represent data from a representative experiment from a total of three conducted. We have also modified the Y-axis label in Figures 3E, 3F, and S4 to % viability, as suggested.

8. Figure 4, error bars for tumour curves and stats for KM curves are missing.

Author response: We thank the reviewer for the constructive recommendation. Just as occurs for patients who enroll in trials, the mice start treatment on different days, whenever their respective tumors reach 4-6 mm in diameter. For this reason, we show the reader both the individual tumor growth curves for each mouse and smoothed growth curves that represent the tumor growth of each group. We rely on the Kaplan-Meier method (Fig. 4C and 4F) to depict survival curves and use Cox proportional hazards to formally assess whether statistical differences exist between treatment groups.

9. Important information about the DSRCT PDX model is missing. How does it compare to the human patient specimen in terms of histology and molecular readout? How many passages from original implantation? Patient information is again missing from this, is this primary specimen, relapse, treatment naïve etc.?

Author response: We thank the reviewer for this important comment. The H&E stained sections demonstrated histological concordance between the patient tumor (PT) and its corresponding patient-matched PDX models (P0-4; **Fig. S10A**). Short tandem repeat (STR) profiling was performed on the PDX to ensure it matched the patient's profile (**Fig. S10B**). The patient specimen used for the PDX came from a white male's primary tumor following neoadjuvant chemotherapy (**Fig. 10C**). The resulting PDX was passaged two times before drug testing.

10. Fig 6A is difficult to see, e.g., early pharmacodynamics changes, e.g., pS6 etc. are they significantly different?

Author response: We have increased the font size of this figure (Fig. 5A in this current revision). The pharmacodynamic changes in pS6 and AKT under AR-ASO treatment are significantly different as compared to the control-ASO treatment group (Supplemental Figure 7F-G).

11. There is an inherent challenge with working with rare entities that good preclinical models are lacking. However, it is also important that any findings reported are robust. It would be critical to see that the data in Figures 6 and 7 are reproduced using a DSRCT PDX model. If the authors are not able to generate short-term primary cultures from the PDX tumors to do these experiments, then I would suggest doing them in PDX tumour explants. It is insufficient to just rely on data from one immortalized cell line to assert conclusions regarding AR biology in DSRCT.

Author response: Per the reviewer's request, we have added Chip-seq analyses of the xenograft and PDX to increase scientific rigor and reproducibility.

Other points

1. This statement is speculation not fact “Though the DSRCT FP shares an N-terminal EWSR1 gene with ES, the tendency to arise within the abdomen must be driven mainly by the aberrant C-terminus WT-1 protein and its downstream epigenetic effects.”

Author response: Most experts believe the WT1 C-terminus fusion partner contributes to DSRCT's unique clinical presentation, which is totally different from Ewing sarcoma. Nevertheless, we could have softened this claim and stated that DSRCT's clinical presentation is *likely* driven by the WT1 and downstream epigenetics. In the end, we have removed this sentence and paragraphs 3-5 entirely for conciseness and readability.

2. Figure S1 only shows a subset of proteins in RPPA, which does not support the statement "unsupervised HC correctly separate normal from DSRCT" as specific proteins were chosen, how were these proteins chosen, please show full datasets.

Author response: We used the Benjamini Hochberg method, with FDR:0.05 and Fold Change ≥ 2 , to select the proteins in Fig S1. Unsupervised hierarchical clustering used the Pearson's centered-distance metric between proteins (columns) and Centroid's linkage clustering method. As requested, the full datasets are provided in the Supplemental Table 5.

3. By definition RPPA is not unbiased, it is a targeted approach because antibody targets are pre-selected, e.g. unlike mass spectrometry. Please amend the text.

Author response: We thank the reviewer for this comment and the text was amended to reflect that the RPPA protein panel is enriched in proteins associated with cancer.

4. Figure 1B, D, please replace biomarkers on x axis with proteins. These are not biomarkers unless independently validated

Author response: We thank the reviewer for this comment, and the Figures 1B and 1D were amended.

5. Figure 2 – consistency of figure and text. Text use high while figure uses positive. What is the cut-off used for IHC and WB

Author response: We thank the reviewer for this comment, the Figure 2 was amended accordingly, and the current Figure reads, high, low, focal, and negative AR expression. The cutoff used for IHC is based the percentage of cells that are positive for AR expression. The patient tumors are either high (>50% cells), low (10-50% cells), focal (1-10% cells), or negative (0-1% cells). The cut-off used for WB is based the relative AR levels across samples. The patient tumors are either high (>10), moderate (between 1 & 10), negative (<1) (Fig. 2D).

6. For PC samples use in Fig 2E-F, what type were they, treatment naïve or have undergone treatment.

Author response: We thank the reviewer for providing this information. The prostate cancer patients involved in the Figure 2E-F are metastatic castrated-resistant (mCRPC). The patients had to be at least 18 years of age and have histologically confirmed prostate carcinoma, with radiographic evidence of metastatic disease. Patients also had to demonstrate tumor progression while on hormone therapy with castrate serum testosterone with biopsy-proven viable disease. The patients must have had a resected prostate cancer mass (primary and/or metastatic site) within 3 months of study entry.

7. Figure 3B stats: It is not appropriate to treat technical replicates as different points for the same patient in a dot blot, should use the mean of the two technical replicates.

Author response: We thank the reviewer for this comment, as suggested by another reviewer of this manuscript, we moved the PSA data to another manuscript.

8. This statement "suggest a novel AR-dependent integrin/NCOA-dependent pathway exists in DSRCT that might trigger DSRCT cell migration and death" – Not supported by integrin data, which they claim is not associated with AR expression? How did they extrapolate this to cell migration and death? Very speculative statement not supported by data

Author response: We thank the reviewer for this comment and we agree that our data doesn't fully support this statement. We have, therefore, amended our text, which now reads, "Though additional samples must be analyzed within a further investigation; these preliminary results suggest a novel AR-dependent integrin/NCOA-dependent pathway exists in DSRCT that might trigger DSRCT cell migration and death."

9. Is Fig 3G all from the same blot? At the moment it looks like it is spliced from a different blot. Again, replicates for blot quant is missing

Author response: We thank the reviewer for requesting more clarification. Yes, Figure 3G was from the same blot, and we had reordered the lanes to place them in a more intuitive order for the reader (e.g., No Rx, Control ASO, then AR-ASO). Our revised manuscript includes an updated Western blot that includes additional replicates in the intended order.

10. RPPA data is not provided/deposited.

Author response: Consistent with standard community practice, our RPPA data has been deposited in GEO (#GSE108687) and will be made publicly available immediately once our manuscript accepted for publication.

REVIEWERS' COMMENTS

Reviewer #1 (Remarks to the Author):

The authors have satisfactorily responded to my comments and concerns.

Reviewer #3 (Remarks to the Author):

The authors have addressed all of my comments in the first review. I just have an additional query that requires some clarification.

In relation to Figure S2D-E, the authors make this statement "Compared to other sarcoma subtypes, most DSRCT samples clustered together based upon their expression of 89 genes associated with the androgen pathway, as defined by KEGG (Fig. S2D). Similarly, DSRCT samples clustered together by their expression of 55 genes linked to canonical androgen signaling (Supplemental Fig. 2F)"

It is well known in multiple published studies that sarcomas separate/cluster by histological subtypes at the gene expression level by unsupervised hierarchical clustering.

1. In order to make the statement "compared to other sarcoma subtypes...", the authors need to unblind Fig S2D-E to show the identify of the other sarcoma subtypes to demonstrate that the other sarcomas do not cluster by subtype. If the other sarcomas similarly cluster together by subtype, then the authors may need to revise some of their statements.

2. Can the authors provide the hierarchical clustering heatmap of the full gene expression dataset to see if DSRCT samples cluster together anyway.

Reviewer #4, Replacement Reviewer for Reviewer #2(Remarks to the Author):

In the manuscript "The Androgen Receptor: A Therapeutic Target in Desmoplastic Small Round Cell Sarcoma", the authors have deciphered the oncogenic potential of androgen receptor (AR) in this rare subtype of sarcoma. The group has taken advantage of the available AR-targeted therapies, namely AR-antagonist enzalutamide and AR anti-sense oligonucleotides for their efficacy in the treatment of AR-positive DSRCT tumors. They have further shown a distinct transcriptional repertoire of AR in DSRCT compared to prostate cancer. Overall, the study addresses an important aspect of the rare sarcoma subtype. Authors have already addressed most of the concerns raised by the previous reviewers. I have rather few queries and some minor comments:

Major comments:

1. It would be of interest to examine the association of AR expression with EWSR1-WT1 fusion transcript in the patients' specimens used for molecular profiling.
2. MACS peaks for figure panel 6E, AR ChIP-Seq data can be shown to represent significant AR enrichment.

Minor comments:

1. Figure 1A, on the x-axis, either remove red/green coding underneath the clustered protein names or use some different colors as that's creating confusion with the scale for protein intensity in the heatmap.
2. Figure 1C, β -actin blot is almost overlapping with the S6-pS235-pS236.
3. Scale bar for the supplementary figures S2A-C is missing.
4. Supplementary figure S2E and S2G, enrichment score and p-value should be provided.
5. Figure 3C, scale bar is not clearly visible.
6. Figure 3D, legend needs to be corrected for the timepoints (0, 5 and 24hrs).
7. In the main text (line 180-181), the authors have mentioned that "Our results suggest that AR-upregulation begins within 5 hours of DHT exposure and peaks in JN-DSRCT, or decreases in

LNCaP cells at 24 hours (Fig. 3C-D)". However, no significant decrease in AR protein levels can be noted at 24 hours in LNCaP cells. Also, the authors can better frame the statement by mentioning "AR-nuclear translocation" instead of AR upregulation as the total AR protein levels with DHT stimulation has not been determined via Western blotting.

8. In the main text (line 186-187), "this antineoplastic effect required 72-hours of DHT pretreatment (Fig. S4)", the statement should be restructured because the current statement symbolizes only treatment with DHT with 72 hours first, and then ASO treatment, however treatment with both (DHT and AR-ASO) was given for 72 hours.

9. Scale bars for Figure 5F and supplementary figures S5A and S5D are not clearly visible.

10. Read signal intensity is not given for the supplementary figures S9G, S11G.

11. In supplementary figures S8D and S10D, the image resolution is poor.

Reviewer #1:

The authors have satisfactorily responded to my comments and concerns.

Reviewer #3:

The authors have addressed all my comments in the first review. I just have an additional query that requires some clarification.

In relation to Figure S2D-E, the authors make this statement "Compared to other sarcoma subtypes, most DSRCT samples clustered together based upon their expression of 89 genes associated with the androgen pathway, as defined by KEGG (Fig. S2D). Similarly, DSRCT samples clustered together by their expression of 55 genes linked to canonical androgen signaling (Supplemental Fig. 2F)". It is well

known in multiple published studies that sarcomas separate/cluster by histological subtypes at the gene expression level by unsupervised hierarchical clustering.

1. To make the statement "compared to other sarcoma subtypes...", the authors need to unblind Fig S2D-E to show the identity of the other sarcoma subtypes to demonstrate that the other sarcomas do not cluster by subtype. If the other sarcomas similarly cluster together by subtype, then the authors may need to revise some of their statements.
2. Can the authors provide the hierarchical clustering heatmap of the full gene expression dataset to see if DSRCT samples cluster together anyway?

Author Response: It's a valid question whether the sarcoma subtypes would cluster together, either by genes associated with the androgen pathway or using the complete set. Accordingly, we produced the unsupervised heatmap below using the 1,500 most differentially expressed genes across all samples. Interestingly, the prostate cancer and DSRCT samples clustered together on a distinct branch apart from the other sarcoma subtypes. DSRCT and prostate cancer appear to share upregulated (A) and downregulated (B) genes that distinguish them from other sarcoma subtypes.

Since this information provides a fuller picture of the similarities and differences between PC, DSRCT, and other sarcoma subtypes, we have added this as Fig. 3 within our revision.

Reviewer #4 (Replacement Reviewer for Reviewer #2):

In the manuscript “The Androgen Receptor: A Therapeutic Target in Desmoplastic Small Round Cell Sarcoma”, the authors have deciphered the oncogenic potential of androgen receptor (AR) in this rare subtype of sarcoma. The group has taken advantage of the available AR-targeted therapies, namely AR-antagonist enzalutamide and AR anti-sense oligonucleotides for their efficacy in the treatment of AR-positive DSRCT tumors. They have further shown a distinct transcriptional repertoire of AR in DSRCT compared to prostate cancer. Overall, the study addresses an important aspect of the rare sarcoma subtype.

Authors have already addressed most of the concerns raised by the previous reviewers. I have rather few queries and some minor comments:

Author Response: We thank Reviewer #4 for his or her additional comments and are happy to address the questions.

Major comments:

1. It would be of interest to examine the association of AR expression with EWSR1-WT1 fusion transcript in the patients’ specimens used for molecular profiling.

Author Response: The reviewer certainly raises a thought-provoking question, that being whether the pathognomonic fusion protein coregulates AR. Though fusion transcripts were detected from RNA-seq data using MapSplice, the expression values lacked the dynamic range needed to reliably correlate with AR expression. In future work, our team hopes to tackle this important question by employing single-nuclei RNA-seq and spatial image -omic analysis (SIO), which would have the added advantage of assessing AR and EWSR1-WT1 in the individual DSRCT lineages (e.g., myogenic, epithelial, and neural).

2. MACS peaks for figure panel 7E, AR ChIP-Seq data can be shown to represent significant AR enrichment.

Author Response: Figure 7E, and all other ChIP-seq figures, were updated to indicate MACS peaks scales.

Minor comments:

1. Figure 1A, on the x-axis, either remove red/green coding underneath the clustered protein names or use some different colors as that’s creating confusion with the scale for protein intensity in the heatmap.

Author Response: As requested, we modified the colors used to label ES and DSRCT samples in Fig. 1A. ES is now consistently color-coded in blue, whereas DSRCT is coded red.

2. Figure 1C, β -actin blot is almost overlapping with the S6-pS235-pS236.

Author Response: Thanks for that tip. We have increased the space between the Western blots in Fig. 1C to avoid overlap between B-actin and S6-pS235-pS236.

3. Scale bar for the supplementary figures S2A-C is missing.

Author Response: We’ve added scale bars to Figures S2A-C.

4. Supplementary figures S2E and S2G, enrichment score and p-value should be provided.

Author Response: As requested, we have added p-values to figures S2E and S2G. The enrichment score can be read directly from the y-axis.

5. Figure 4C, the scale bar is not clearly visible.

Author Response: Good point. We've revised this image using a larger scale bar in the top leftmost image.

6. Figure 4D, legend needs to be corrected for the time points (0, 5, and 24hrs).

Author Response: Thanks for catching that mistake in our Fig. 4D legend, which was revised to specify the 0, 5, and 24-hour time points.

7. In the main text (line 180-181), the authors have mentioned that "Our results suggest that AR-upregulation begins within 5 hours of DHT exposure and peaks in JN-DSRCT, or decreases in LNCaP cells at 24 hours (Fig. 4C-D)". However, no significant decrease in AR protein levels can be noted at 24 hours in LNCaP cells. Also, the authors can better frame the statement by mentioning "AR-nuclear translocation" instead of AR upregulation as the total AR protein levels with DHT stimulation has not been determined via Western blotting.

Author Response: Very good point. We have amended the manuscript's wording per the reviewer's suggestion. That clause now reads, "Our results suggest that AR-nuclear translocation begins within 5 hours of DHT exposure and peaks by 24 hours (Fig. 4C-D)".

8. In the main text (line 186-187), "this antineoplastic effect required 72-hours of DHT pretreatment (Fig. S4)", the statement should be restructured because the current statement symbolizes only treatment with DHT with 72 hours first, and then ASO treatment, however treatment with both (DHT and AR-ASO) was given for 72 hours.

Author Response: We have clarified this in the main text, which now reads "Notably, this antineoplastic effect required concurrent administration with DHT (Fig. S4)."

9. Scale bars for Figure 6F and supplementary figures S5A and S5D are not clearly visible.

Author Response: We have increased the size of the scale bars in Fig. 6F, S5A, and S5D images, as requested.

10. Read signal intensity is not given for the supplementary figures S9G, S11G.

Author Response: At the reviewer's suggestion, we have added the read signal intensity scales to any ChIP-seq images where they were missing. Additionally, we increased the font size to improve legibility of the intensity scale and MACS peaks numbers.

11. In supplementary figures S8D and S10D, the image resolution is poor.

Author Response: Thanks for catching this. We have updated S8D and S10D with the original high-resolution images.